# Rolling controls sperm navigation in response to the dynamic rheological properties of the environment

Meisam Zaferani, Farhad Javi, Amir Mokhtare, Peilong Li, Alireza Abbaspourrad*

Department of Food Science, College of Agriculture and Life Sciences, Cornell University, Ithaca, United States

**Abstract** Mammalian sperm rolling around their longitudinal axes is a long-observed component of motility, but its function in the fertilization process, and more specifically in sperm migration within the female reproductive tract, remains elusive. While investigating bovine sperm motion under simple shear flow and in a quiescent microfluidic reservoir and developing theoretical and computational models, we found that rolling regulates sperm navigation in response to the rheological properties of the sperm environment. In other words, rolling enables a sperm to swim progressively even if the flagellum beats asymmetrically. Therefore, a rolling sperm swims stably along the nearby walls (wall-dependent navigation) and efficiently upstream under an external fluid flow (rheotaxis). By contrast, an increase in ambient viscosity and viscoelasticity suppresses rolling, consequently, non-rolling sperm are less susceptible to nearby walls and external fluid flow and swim in two-dimensional diffusive circular paths (surface exploration). This surface exploration mode of swimming is caused by the intrinsic asymmetry in flagellar beating such that the curvature of a sperm's circular path is proportional to the level of asymmetry. We found that the suppression of rolling is reversible and occurs in sperm with lower asymmetry in their beating pattern at higher ambient viscosity and viscoelasticity. Consequently, the rolling component of motility may function as a regulatory tool allowing sperm to navigate according to the rheological properties of the functional region within the female reproductive tract.

*For correspondence:
alireza@cornell.edu

Competing interests: The authors declare that no competing interests exist.

## Introduction

In mammals, sperm must migrate through the female reproductive tract to fertilize an egg (*Suarez and Pacey, 2006*; *Suarez, 2016*). During this migration, sperm require navigational mechanisms to swim in the correct direction (*Eisenbach and Giojalas, 2006*; *Kaupp et al., 2008*). These navigational mechanisms rely on external and dynamic biochemical and biophysical cues that are present in the female reproductive tract (*Kaupp et al., 2008*; *Bahat et al., 2003*; *Tung et al., 2015a*). Although the role of biochemical cues in mammalian sperm navigation remains poorly understood (*Kaupp et al., 2008*; *Suarez, 2008*), in vitro and in vivo studies have provided evidence for two navigational mechanisms that rely on external biophysical cues, namely rheotaxis (*Miki and Clapham, 2013*; *Kantsler et al., 2014*; *Bukatin et al., 2015*; *Tung et al., 2015b*) and wall-dependent navigation (*Guidobaldi et al., 2014*; *Nosrati et al., 2016*; *Denissenko et al., 2012*; *Zaferani et al., 2019*; *Wang and Larina, 2018*). Rheotaxis, as an upstream swimming in response to an external fluid flow, has been observed and quantitatively studied for bovine, human, and mouse sperm (*Miki and Clapham, 2013*; *Kantsler et al., 2014*). Wall-dependent navigation, as sperm response to the nearby physical boundaries such as walls of the female reproductive tract, has also been observed and characterized for bovine and human sperm (*Tung et al., 2015a*; *Denissenko et al., 2012*).

Although wall-dependent navigation combined with rheotaxis may characterize regulatory mechanisms of sperm navigation within the complex geometry of the female reproductive tract and under dynamic fluid flow, it remains unclear how sperm rolling contributes to these navigational mechanisms (*Gadadhar et al., 2021*; *Miller et al., 2018*; *Drake, 1974*; *Babcock et al., 2014*; *Schiffer et al., 2020*). Furthermore, the dynamic biophysical factors of the female reproductive tract are not limited to its complex geometry and the varying flow of the mucus within it. The rheological properties of the mucus also change according to the functional region within the female reproductive tract (*Suarez and Pacey, 2006*; *Carlstedt and Sheehan, 1984*; *Tung et al., 2017*). It also remains poorly understood how the rheology of the environment influences sperm swimming behavior, and in particular, rheotaxis and wall-dependent navigation.

To address these questions, we investigated bovine sperm motion in a microfluidic device to identify how the rolling component of its motility contributes to navigation under simple shear flow and within a quiescent reservoir. To avoid errors arising from studying sperm motility under external fluid flow, such as experimental inaccuracies caused by decoupling the effect of flow on sperm motion from active swimming, we employed a multi-step approach. In this approach, we first isolated sperm within a quiescent reservoir using a rheotaxis-based method to ensure that sperm within the reservoir could swim upstream before entering the reservoir (*Zaferani et al., 2018*). We then characterized the components of sperm motility, including flagellar beating and rolling, in viscous and viscoelastic media, in the absence of external flow. Finally, we studied wall-dependent navigation within the reservoir and, by tracking the sperm prior to their entry into the reservoir, we evaluated rheotaxis to identify the function of rolling in sperm navigation under fluid flow.

We found that rolling enables sperm to swim progressively even when the sperm flagellar beating pattern is intrinsically asymmetric, which subsequently promotes rheotaxis and wall-dependent navigation. Sperm that lack rolling swim along two-dimensional (2D) diffusive circular paths and are less susceptible to being influenced by the nearby walls and external flow. We observed that the suppression of rolling occurs by increasing ambient viscosity or viscoelasticity, such that an increase in ambient rheological properties transitions progressive motion into 2D diffusive circular surface exploration. Such diffusive circular surface exploration is caused by the intrinsic asymmetry of flagellation and the curvature of the circular path is proportional to the level of asymmetry. We noticed that suppression of rolling was reversible, as a decrease in ambient viscosity or viscoelasticity resulted in reactivation of rolling. Furthermore, we found out that the level of flagellar asymmetry in a sperm population forms a continuum, and suppression of rolling in sperm with lower flagellar asymmetry occurs at higher viscosity or viscoelasticity.

Sperm swimming behavior transitions between progressive and diffusive circular motions after each incidence of suppression or reactivation of rolling. Since the suppression or reactivation of rolling relies on changes in the viscosity or viscoelasticity of the media, sperm swimming behavior manifests differently in response to the rheological properties of the environment. Because the characteristics of these swimming behaviors (circular versus progressive) are different and the viscosity and viscoelasticity of the mucus in the female reproductive tract varies according to functional regions (*Suarez and Pacey, 2006*; *Tung et al., 2017*; *Ishimoto and Gaffney, 2018*; *Ishimoto and Gaffney, 2016*; *Gaffney et al., 2011*), our results suggest that rolling potentially enables sperm to regulate its navigation in response to the dynamic rheological properties of the mucus in the tract.

## Results

To select sperm based on their rheotactic behavior and to isolate them inside the quiescent reservoir, we used a microfluidic corral system. Within the reservoir filled with standard Tyrode's albumin lactate pyruvate medium (TALP), a sub-population of sperm (<5%) exhibited an in-plane 2D asymmetric flagellar beating pattern, where the midpiece of each flagellum was consistently bent more significantly to one side (*Figure 1A* and *Video 1*). This asymmetric beating pattern results in circular swimming at an angular velocity of $\Omega$, whereas other sperm within the population swim progressively (*Figure 1B*). Unlike circular motion, progressive motion is not just produced by 2D in-plane symmetric flagellation, rather frequent but irregular rollings contribute to sperm motility (*Figure 1C* and *Video 2*). The rolling component was detected when a change in the light intensity of the sperm heads was visualized under a phase contrast microscope (*Figure 1C*). Rollings occur rapidly and discontinuously (*Drake, 1974*; *Babcock et al., 2014*), such that the elapsed time between two

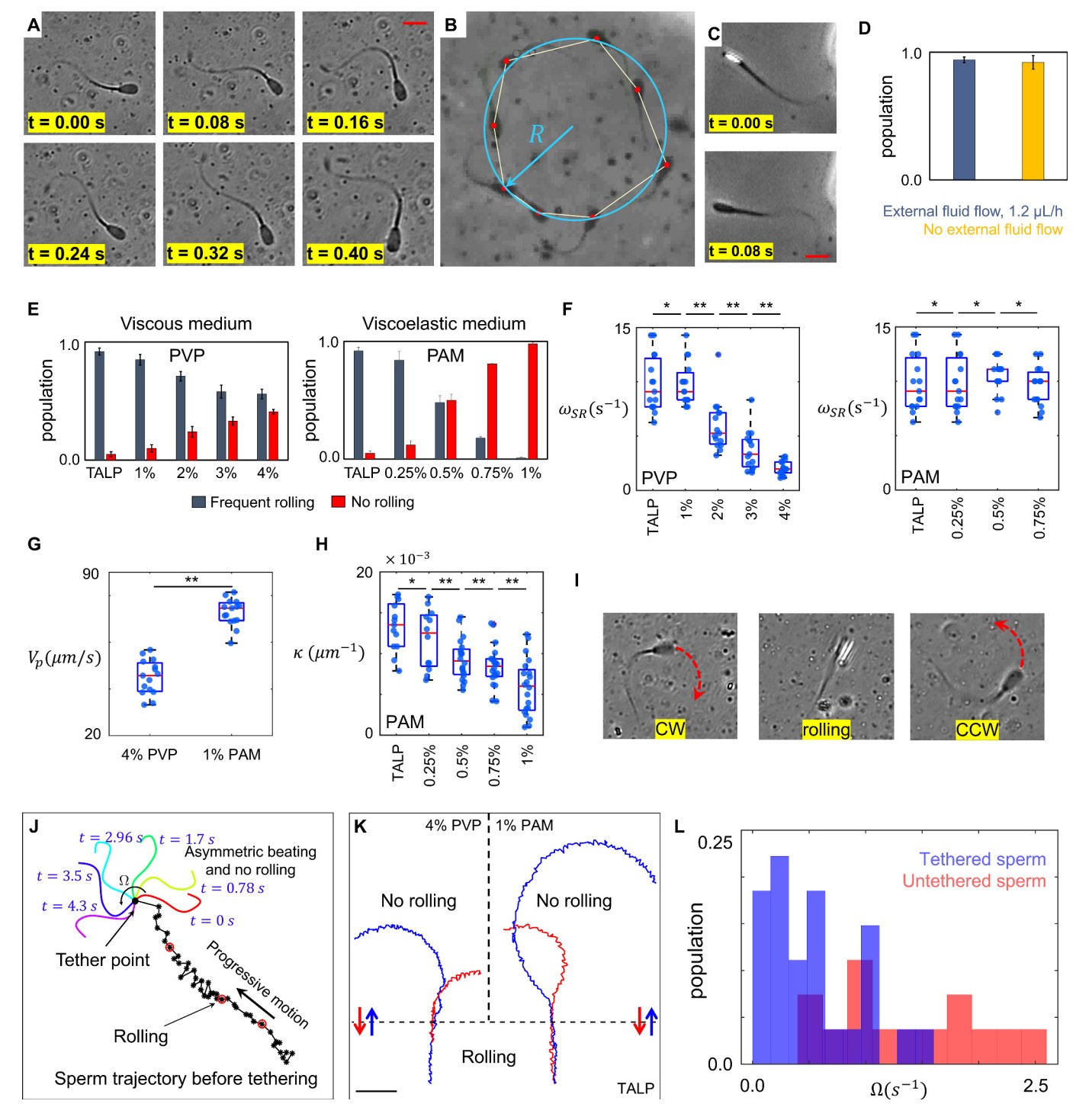

**Figure 1.** Sperm motility within the quiescent reservoir. (**A**) Flagellar asymmetric beating. The midpiece of the flagellum consistently bends more prominently in one direction. (**B**) The circular motion caused by the asymmetric beating. A least-squares fitting algorithm was used to fit a circle into the sperm trajectory. (**C**) Rolling under a phase-contrast microscope. (**D**) All progressively motile sperm exhibit frequent rolling under flow and within the quiescent reservoir. (**E**) Increasing viscosity and viscoelasticity suppresses rolling and increases circular motion within the population (p>0.05). (**F**) An increase in viscosity results in a decrease in rolling frequency in rolling sperm while an increase in viscoelasticity does not change the frequency of rolling in rolling sperm. (**G**) Propulsive velocity of non-rolling sperm in 4% polyvinylpyrrolidone (PVP) is significantly lower than that of 1% polyacrylamide (PAM). (**H**) Suppression of rolling for sperm exhibiting less asymmetry in their flagellation occurs at higher viscoelasticity. (**I**) Infrequent rolling changes the direction of motion from CW to CCW or vice versa. (**J**) Sperm rotating around the tethering point upon adhesion to the glass surface. Progressive

*Figure 1 continued on next page*

*Figure 1 continued*

motion before tethering was generated by frequent rolling. (**K**) Reversible transition between progressive and circular motions through suppression and reactivation of rolling. Trajectories in Tyrode's albumin lactate pyruvate medium (TALP) and 1% PAM were obtained with 0.08 s intervals. Intervals in 4% PVP were 0.16 s. (**L**) Distribution of sperm angular velocity $\Omega$ for both tethered and untethered sperm. **p>0.05, **p<0.01. The p-values were obtained from two-tailed t-tests, with adjustments for multiple comparisons (Bonferroni correction). The concentrations are reported in weight percent.

The online version of this article includes the following figure supplement(s) for figure 1:

**Figure supplement 1.** The rheological characteristics of polyvinylpyrrolidone (PVP)- and polyacrylamide (PAM)-based sperm media.

---

consecutive rolling events $(T_{SR})$ is not constant. While tracking the sperm within the reservoir and under the flow prior to their entry into the reservoir, we found that progressive motility included the rolling component, both under the flow and within the quiescent reservoir, whereas sperm exhibiting a circular motion did not exhibit rolling under either condition (*Figure 1D*). Although rolling occurs independently of external fluid flow, it does depend on the rheological properties of the medium, as the percentage of rolling sperm decreases with an increase in the viscosity and viscoelasticity of the medium (*Figure 1E*, *Video 3*). Our results indicate that the suppression of rolling is more sensitive to viscoelasticity than to viscosity. We used varying concentrations of polyvinylpyrrolidone (PVP) and polyacrylamide (PAM) to prepare viscous and viscoelastic solutions (*Tung et al., 2017*), the rheological characteristics of which are presented in Section I of Appendix 1.

We observed that, in all the solutions we used, rolling sperm swam progressively, whereas non-rolling sperm exhibited 2D asymmetric flagellar beatings and swam in circles, such that the curvature of the circular path was found to be proportional to the asymmetry level of flagellar beating. Furthermore, an increase in viscosity decreased the frequency of rolling $\left(\omega_{SR} = \frac{1}{T_{SR}}\right)$ in rolling sperm (*Figure 1F*) as well as their progressive velocities, such that rolling sperm continue to swim progressively but slowly, with rolling occurring at lower frequency in a viscous solution. An increase in viscoelasticity did not, however, change the frequency of rolling and the average path velocity of rolling sperm (*Figure 1F*). We also observed that suppression of rolling in the viscous solution (4% PVP) occurred with a significant decrease in the propulsive velocity of non-rolling sperm, whereas rolling suppression in the viscoelastic solution (1% PAM) did not result in a decrease in the propulsive velocity of non-rolling sperm (*Figure 1G*). Considering that viscosity of 1% PAM solution is an order of magnitude higher than that of 4% PVP solution (Appendix I, Section I), these results seem to contradict our claim that an increase of viscosity leads to suppression of rolling as well as decrease in the propulsive velocity. However, we stress that the storage modulus $(G')$ of 1% PAM solution which

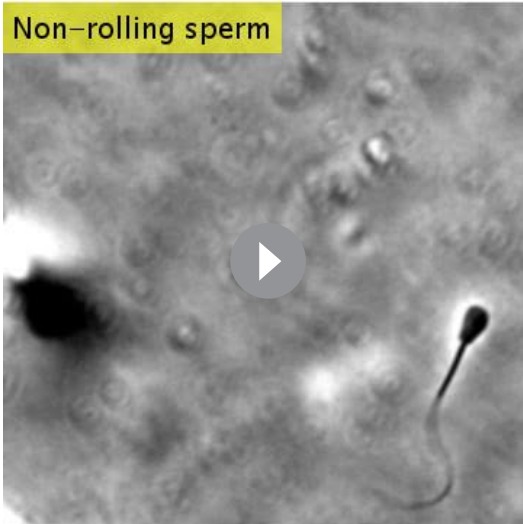

**Video 1.** Sperm in-plane two-dimensional (2D) asymmetric flagellar beating results in a circular motion.
https://elifesciences.org/articles/68693#video1

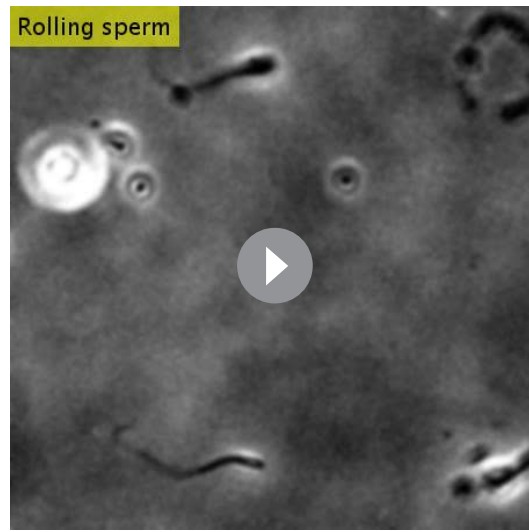

**Video 2.** Rolling sperm with progressive swimming behavior.
https://elifesciences.org/articles/68693#video2

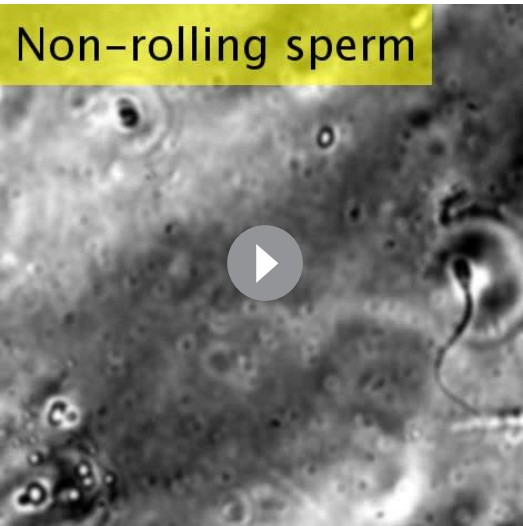

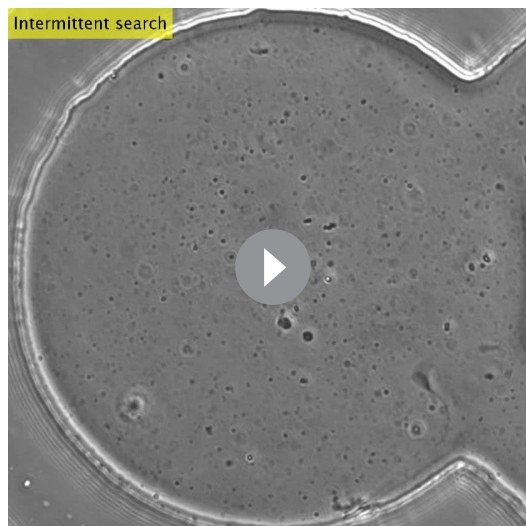

**Video 3.** Suppression of rolling in the viscoelastic solution. https://elifesciences.org/articles/68693#video3

**Video 4.** Sperm infrequent rolling and intermittent search. https://elifesciences.org/articles/68693#video4

represents the elastic properties of the fluid was two orders of magnitude higher than that of 4% PVP solution (SI, Section I). The higher elasticity of 1% PAM indicates that this increase contributed significantly to the suppression of rolling, while it did not result in decrease of the sperm propulsive velocity.

We also noticed that the range of curvature ($\kappa$) of the circular path in the non-rolling sperm population depended on ambient viscoelasticity (*Figure 1H*). As shown in *Figure 1H*, the suppression of rolling in sperm with a higher degree of asymmetry in their beating pattern occurred at lower ambient viscoelasticity, whereas higher ambient viscoelasticity was needed to suppress the rolling of sperm that exhibited lower asymmetry in their beating patterns. Similar behavior was observed when we increased the viscosity of the solution.

In TALP + 1% and 2% PVP solutions, some sperm exhibiting circular motion (<10%) also exhibited infrequent rollings, such that $\Omega T_{SR} > 2\pi$. Such infrequent rollings changed the direction of the circular motion (*Figure 1I* and *Video 4*) and resulted in abrupt relocations of the circular path's center without significantly changing the curvature of the path.

Although our results indicate that rolling is a key contributor to progressive motility, more evidence is needed to validate the hypothesis. Therefore, we decreased the concentration of bovine serum albumin to 0.5% in TALP and tethered sperm heads to the glass surface upon their entry into the reservoir (*Saggiorato et al., 2017*). Tethering the sperm heads to the glass surface suppressed the rolling component, and the 2D flagellar beating pattern was observed separately from rolling. We observed that, without rolling, flagellation was not necessarily symmetric and >80% of the sperm began rotating around the sperm head upon tethering, although their motion was progressive prior to the surface binding (*Figure 1J*, *Videos 5* and *6*).

Another piece of evidence that validates our hypothesis was observed by tracking single sperm migrating between TALP and 1% PAM or 4% PVP solutions (*Figure 1K*). The representative

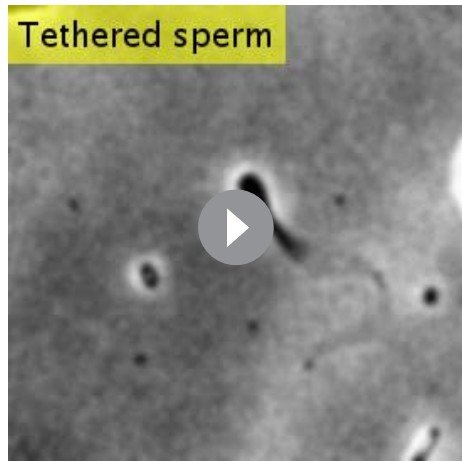

**Video 5.** Tethered sperm with asymmetric flagellar beating. https://elifesciences.org/articles/68693#video5

sperm trajectories shown in *Figure 1K* indicate that suppression of rolling upon exiting TALP and entering into the viscous or viscoelastic solutions (blue trajectories) resulted in an instantaneous transition in sperm swimming behavior from progressive to circular. The transition in sperm swimming behavior was found to be reversible, as reactivation of rolling upon exiting the viscous or viscoelastic solutions and entering into TALP (red trajectories) caused sperm to start swimming progressively. This reversible transition was observed in more than 95% of sperm cells (total count = 109) migrating between TALP and 1% PAM or 4% PVP solutions. This finding agrees with the results shown in *Figure 1G*, as sperm velocity of average path for the blue and red trajectories was measured to be $(70 \pm 5)$ µms$^{-1}$ in TALP, $(45 \pm 5)$ µms$^{-1}$ in 4% PVP, and $(70 \pm 5)$ µms$^{-1}$ in 1% PAM solutions.

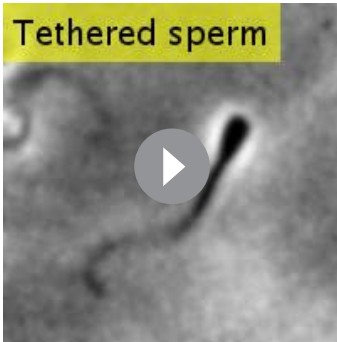

**Video 6.** Tethered sperm with symmetric flagellar beating.
https://elifesciences.org/articles/68693#video6

We measured the angular velocities of all tethered sperm and compared them with that of untethered sperm, which were swimming freely in circles (*Figure 1L*). Consistent with *Figure 1H*, tethered sperm exhibited lower angular velocities than untethered sperm. Thus, rolling sperm exhibit lower asymmetry than their non-rolling counterparts and, accordingly, higher ambient viscosity and viscoelasticity is needed to suppress the rolling motion of sperm with lower asymmetry in flagellation and vice versa.

## Rolling and progressive motion

To quantify the relationship between rolling and progressive motion, we first characterized asymmetry in the flagellar beating pattern. In agreement with *Friedrich et al., 2010*, measuring bending in the midpiece of the flagellum in time (*Figure 2A*) and the corresponding normalized fast Fourier transform $(P^*(t))$ revealed the presence of a zeroth harmonic within the frequency domain of flagellation (*Figure 2B*). Note that the constant offset in the normalized power spectrum is white noise in our measurement system. Furthermore, the experimental noise coming from our measurement system (e.g., image processing) is also included in the peak width around the first harmonic frequency. That is, the peak width centered at $\omega$ includes the intrinsic noise originated from flagellar sources, as well as the noise associated with our measurement system.

Writing the flagellar beating pattern in Fourier series ansatz (*Equation 1*) and further modeling the flagellar beating pattern (SI, Section II), we found that $\sigma = \frac{a_1 - a_0}{a_1}$ approximately presents the asymmetry level in the beating pattern. That is, $\sigma = 1$ presents symmetric beating whereas lower values correspond to higher asymmetry in the beating pattern:

$$y(x,t) = \sum_{n=0} a_n \cos(n\omega t - kx). \tag{1}$$

In *Equation 1*, the $x$ axis was set parallel to the flagellum at its straight-line form, $k = \frac{2\pi}{\lambda}$ (with $\lambda \approx L$) is the wave number, $\omega$ is the main frequency, and $a_n$ is the amplitude of the $n$th harmonic. Note that the amplitude and frequencies of beating include delta-correlated Gaussian noise that can be described as $a_n = \tilde{a}_n(1 + \eta_n(t))$ and $\omega_n = \tilde{\omega}_n(1 + \xi(t))$, respectively, with $\langle \eta_n(t) \rangle = \langle \xi(t) \rangle = 0$, $\langle \eta_n(t)\eta_{n'}(t') \rangle = D_n \delta_{nn'} \delta(t - t')$, and $\langle \xi(t)\xi(t') \rangle = D_\omega \delta(t - t')$. The tilde sign above a quantity represents its time average value. Following *Saggiorato et al., 2017*; *Friedrich et al., 2010*; *Elgeti et al., 2015*, we applied small amplitude (i.e., $\forall n \to a_n \ll L$) and length preservation constraints and used resistive force theory (SI, Sections II–IV), to find that the zeroth harmonic in the beating pattern yields a toque as follows:

$$\tau_f = (\xi_N - \xi_T)\omega k L a_0 \sum_{n=0} n\tilde{a}_n^2. \tag{2}$$

Note that $\xi_T$ and $\xi_N$ are drag coefficients in the tangential and normal directions, respectively,

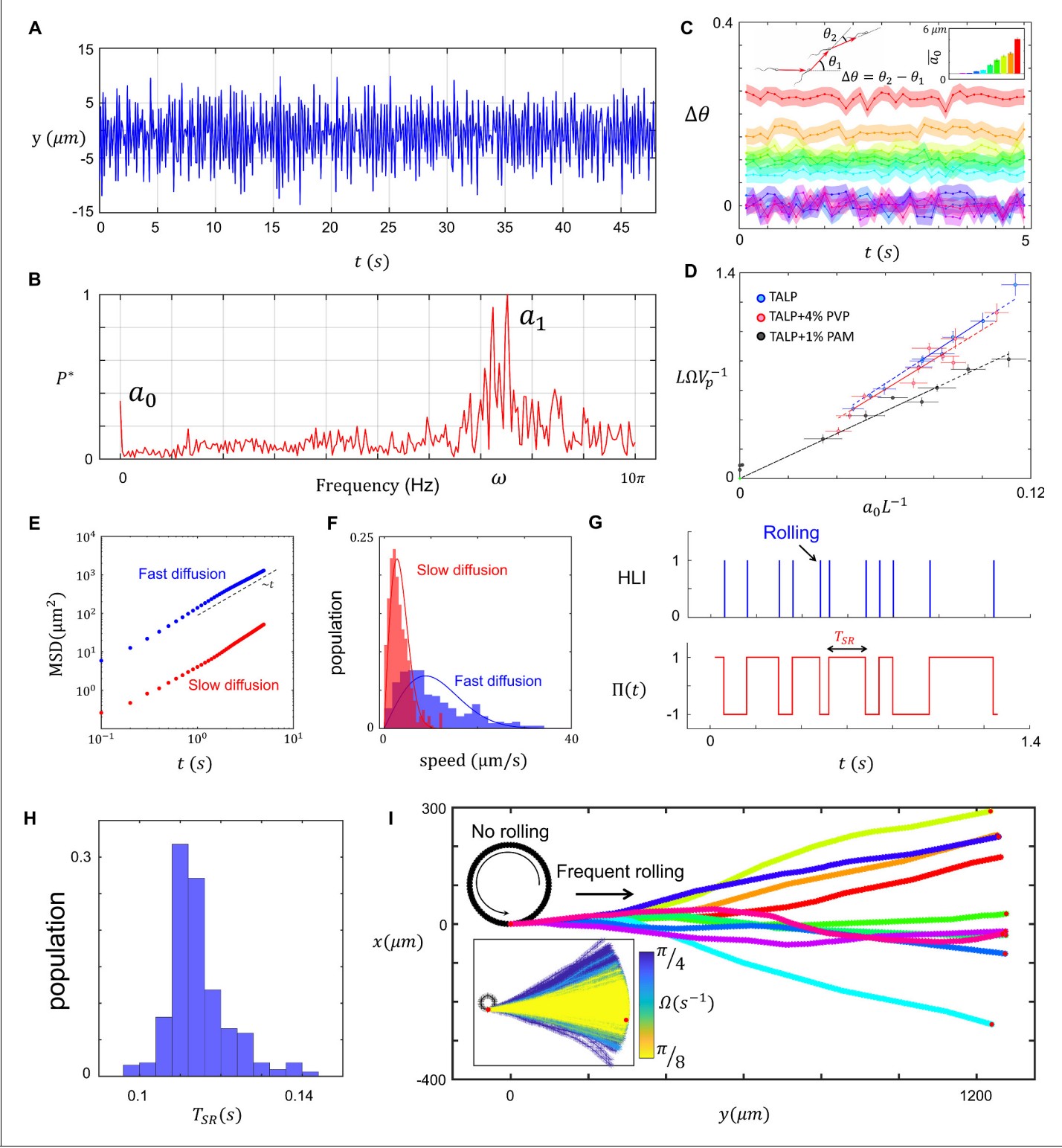

**Figure 2.** Characteristics of sperm motility. (**A**) Bending in the midpiece ($y(t)$). (**B**) The Fourier transform of $y(t)$, $P^*(\omega)$. (**C**) The average angle that the sperm sweeps in each beat, $\Delta\theta$ and the corresponding $\bar{a}_0$. (**D**) The normalized curvature of the path $\left(L\Omega V_p^{-1}\right)$ versus the normalized amplitude of the zeroth harmonic ($a_0 L^{-1}$) measured in Tyrode's albumin lactate pyruvate medium (TALP), TALP + 4% polyvinylpyrrolidone (PVP), and TALP + 1% polyacrylamide (PAM). (**E**) Mean square displacement of the circular path's center and (**F**) distribution of the center's speed for two sperm swimming in circles. (**G**) The plot of head light intensity (HLI) versus time and the corresponding $\Pi(t)$. (**H**) The distribution of $T_{SR}$ for a single progressively swimming

*Figure 2 continued on next page*

*Figure 2 continued*

sperm with rolling. The distribution has mean and standard deviation of $\mu_{SR}$ and $\sigma_{SR}$, respectively. (I) A single sperm with arbitrary $\Omega$ and frequent rollings swims progressively. The direction of progressive motion has a similar distribution to that of the frequent rolling, with mean and standard deviation of 0 and $\sqrt{n_{SR}}\sigma_{SR}\tilde{\Omega}$, respectively. See the effect of $\Omega$ in the inset plot.

The online version of this article includes the following figure supplement(s) for figure 2:

**Figure supplement 1.** The beating pattern of the sperm flagellum.

**Figure supplement 2.** The harmonics of sperm beating.

**Figure supplement 3.** Schematic demonstrating the variables used in *Equations S14–S17*.

**Figure supplement 4.** Dependence of sperm trajectory on $\widetilde{\Omega T_{SR}}$.

**Figure supplement 5.** The influence of frequent rolling on sperm rheotaxis.

**Figure supplement 6.** Experimental tracking of the sperm under an external fluid flow.

and $L$ is the sperm length. Although the propulsive force produced by the flagellum correlates with the characteristics of the first and higher harmonics, the amplitude of the zeroth harmonic is involved solely in the torque produced (*Equation 2*). Applying the zero net torque and force constraints, we found that propulsive $\left(\tilde{V}_P\right)$ and angular $\left(\tilde{\Omega}\right)$ velocities were related through $a_0$:

$$\tilde{\Omega} \propto \frac{\xi_T}{\xi_N}\left(\frac{a_0}{L^2}\tilde{V}_P\right). \tag{3}$$

To verify *Equation 3*, we measured the angular velocity of the sperm and plotted a normalized path curvature $\left(L\Omega V_P^{-1}\right)$ with respect to the normalized amplitude of the zeroth harmonic ($a_0 L^{-1}$) in TALP, TALP + 4% PVP (viscous), and TALP + 1% PAM (viscoelastic) solutions (*Figure 2C and D*). The linear correlation between $L\Omega V_P^{-1}$ and $a_0 L^{-1}$ is consistent with our mathematical arguments (*Equations 1–3*), confirming that $a_0$ modulates the curvature of the non-rolling sperm path. Furthermore, the linear relationship between $L\Omega V_P^{-1}$ and $a_0 L^{-1}$ predicted by the resistive force theory is preserved for sperm motion in a viscoelastic solution whereas the slope differs from that observed for sperm motion in standard and viscous solutions (*Zhang and Goldman, 2014*; *Teran et al., 2010*; *Li and Ardekani, 2015*; *Riley and Lauga, 2017*; *Spagnolie et al., 2013*).

The resistive force theory, as a mean field approach, cannot explain the observed inconsistency in the circular path. The inconsistency in the circular path, however, can be characterized by quantifying the random fluctuations at the center of the circular path (*Ma et al., 2014*). We noticed that fluctuations at the center, and thus circular motion, are diffusive in character as the mean square displacement (MSD) of the center is proportional to the elapsed time: $\mathrm{MSD} \sim t$ (*Figure 2E*). This diffusive motion can be quantified by the diffusion coefficient of the center determined by the intercept of MSD, or alternatively, the center's speed distribution (*Figure 2F*).

We measured the head light intensity (HLI) of the sperm over time to characterize rollings during progressive motility (*Figure 2G*). HLI is a pulse-type quantity, with the pulse duration shorter than the time that elapses between two consecutive pulses, $T_{SR}$. Therefore, we defined the edge-sensitive function $\Pi(t)$ with $\Pi(0) = 1$, such that $\Pi(t)$ is multiplied by $-1$ at each positive edge of HLI (*Figure 2G*). Note that $\Pi(t)$ captures the rolling component as a rapid switch in the direction of asymmetry (*Figure 1I*); rolling can therefore be incorporated into *Equation 1* with $\Pi(t)$:

$$y(x,t) = \sum_{n=0} \Pi(t) a_n \cos(n\omega t - kx). \tag{4}$$

Solving the equations of motion using *Equation 4*, we found that, depending on $\widetilde{\Omega T_{SR}}$, sperm swim along varying pathways with average progressive velocity of:

$$\bar{V} = \tilde{V}_P \frac{2\sin\left(\frac{\widetilde{\Omega T_{SR}}}{2}\right)}{\widetilde{\Omega T_{SR}}} \tag{5}$$

(*Figure 2—figure supplement 4A and B*). At the frequent rolling limit $\left(\widetilde{\Omega T_{SR}} \to 0\right)$, $\bar{V}$ approaches $\tilde{V}_P$, which means that frequent rolling asymptotically yields progressive motion with the average

path velocity equal to the propulsive velocity even though $\tilde{\Omega} \neq 0$. Our experimental measurements are consistent with this finding because $\widetilde{\Omega T_{SR}}$ is less than $2\pi$ for rolling sperm, whereas for circular motion with infrequent rolling it is greater than $2\pi$ (**Figure 2—figure supplement 4C**). Note that for non-rolling sperm, $\widetilde{\Omega T_{SR}} \to \infty$.

Even though $\widetilde{\Omega T_{SR}}$ determines the average path, how do deviations of $T_{SR}$ from the mean during motion influence the trajectory? Our measurements of $T_{SR}$ for one rolling sperm for ~40 s indicate that the mean and standard deviation of this random variable are $\tilde{T}_{SR} \approx 0.11$ s, $\sigma_{SR} \approx 0.02$ s, respectively (**Figure 2H**). Combining arbitrary values for $\tilde{\Omega}$ with $T_{SR}$ distribution, we found that the direction of the average path obeys a similar distribution with mean of 0 and standard deviation of $\sqrt{n_{SR}}\sigma_{SR}\tilde{\Omega}$ (**Figure 2I**), in which $n_{SR}$ is the number of rolling occurrences.

Considering rolling as a switch in the direction of asymmetry, we investigated how rolling influenced sperm rheotaxis. The sperm-rheotactic behavior and angular velocity of upstream orientation is modeled using an Adler-type **Equation 10**:

$$\Omega_{RH} = -A\gamma\sin\theta, \tag{6}$$

in which $\gamma$ is the shear rate, $A$ is a constant, and $\theta$ is the relative orientation of the sperm with respect to the external fluid stream. For sperm with intrinsic angular velocity $(\Omega)$, the net angular velocity under fluid flow is $\Omega_{RH} + \Omega$, which orients the sperm with respect to the flow during upstream motion $(\theta_{UP})$:

$$\theta_{UP} = \sin^{-1}\left(\frac{\Omega}{A\gamma}\right). \tag{7}$$

Including the rolling component in **Equation 7** using $\Pi(t)$, the average orientation of the sperm with respect to flow during upstream motion is:

$$\tilde{\theta}_{UP} = \left\langle \sin^{-1}\left(\frac{\Omega\Pi(t)}{A\gamma}\right) \right\rangle = \widetilde{\Pi(t)}\sin^{-1}\left(\frac{\Omega}{A\gamma}\right) \ll \sin^{-1}\left(\frac{\Omega}{A\gamma}\right). \tag{8}$$

Notably, $\widetilde{\Pi(t)} \ll 1$. **Equation 8** predicts that rollings result in significant decay in $\tilde{\theta}_{UP}$. Consequently, **Equations 9 and 10** respectively present sperm net velocity in the upstream direction $(\tilde{V}_{UP})$ in the presence or absence of frequent rollings:

$$\tilde{V}_{UP} = V_P\cos(\tilde{\theta}_{UP}) - V_F \approx V_P - V_F \tag{9}$$

$$\tilde{V}_{UP} \approx V_P - V_F - \left(V_N\frac{\Omega}{A\gamma} + \frac{1}{2}V_P\left(\frac{\Omega}{A\gamma}\right)^2\right). \tag{10}$$

In **Equations 9 and 10**, $V_P$ and $V_N$ are sperm propulsive and perpendicular velocities (corresponding to the angular velocity), respectively, whereas $V_F$ is the external flow velocity (SI, Section V). To experimentally confirm the prediction obtained from **Equations 9 and 10**, we back-tracked and analyzed sperm motion under flow prior to their entry into the quiescent zone. We observed that $\tilde{\theta}_{UP}$ is greater for non-rolling sperm $(\tilde{\theta}_{UP} = 40° \pm 10°)$ than for those exhibiting the rolling motion $(\tilde{\theta}_{UP} = 10° \pm 5°)$. Furthermore, the average upstream velocity of rolling sperm was much higher $(\tilde{V}_{UP} = 60 \pm 10 \,\mu\text{m/s})$ than that of non-rolling sperm $(\tilde{V}_{UP} = 30 \pm 10 \,\mu\text{m/s})$ (SI, Section V). These results indicate that, even though the rolling component is not required for sperm rheotaxis, it facilitates rheotactic behavior by minimizing the angle between sperm orientation and the external fluid flow, thus maximizing the upstream component of their motion.

## Sperm surface exploration

Non-rolling sperm swim along diffusive circular paths and explore the surface (**Ma et al., 2014**). To characterize the diffusivity in circular motion (**Figure 2E and F**), we overlayed consecutive images of one sperm with 0.08 s intervals over different times (**Figure 3A**). The thickness of the combined circular paths $(\delta_{1-4})$ increases over time in all directions, $\delta_{1-4} \propto \sqrt{t}$ (**Figure 3B**) as predicted by the

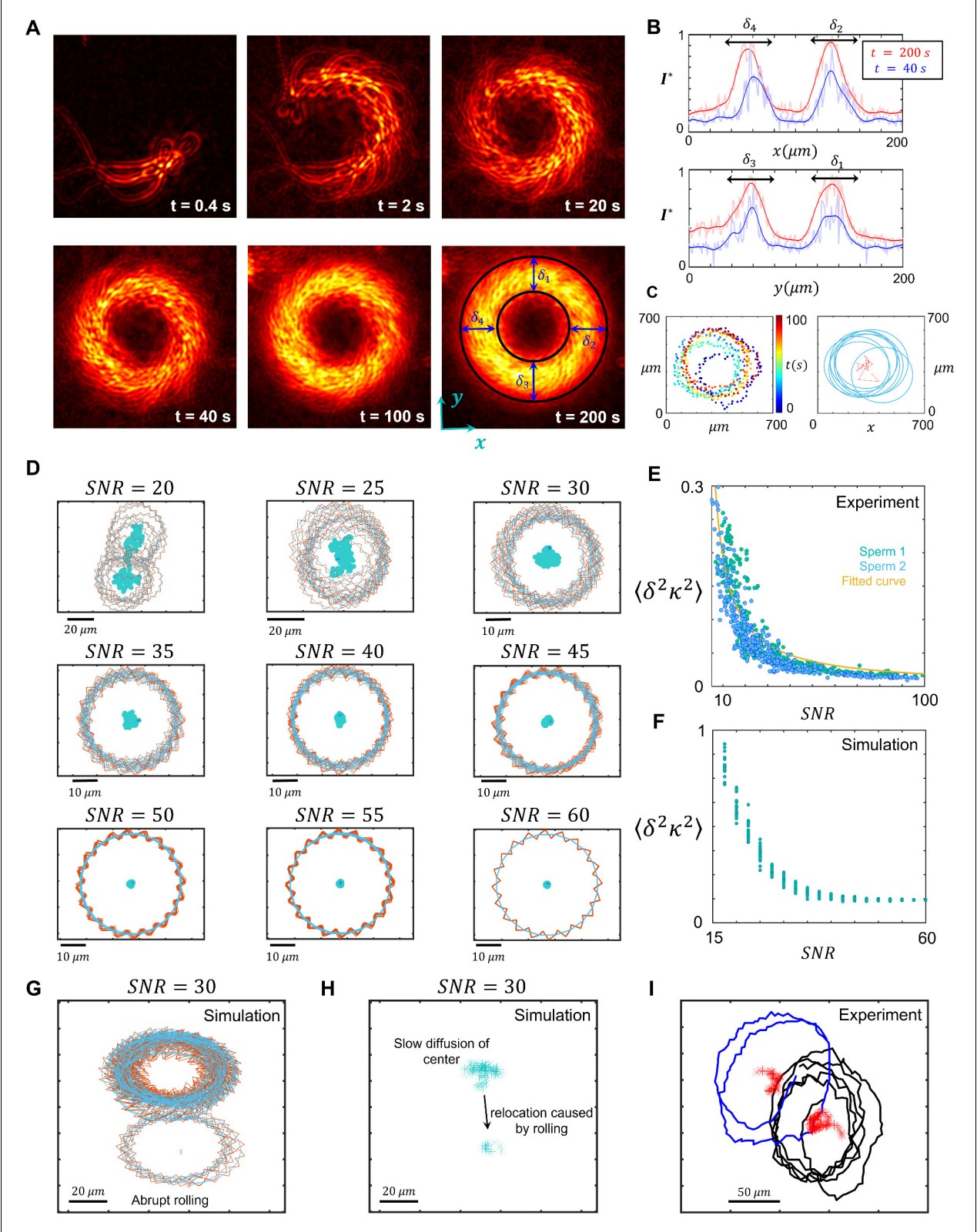

**Figure 3.** Infrequent rolling and diffusive circular motion. (**A**) Overlayed consecutive images of a single sperm. Intervals: 0.08 s. (**B**) Diffusive motion increases $\delta_{1-4}$ in time. (**C**) Manual tracking of the sperm and the corresponding circular path's center. (**D**) Trajectories of the sperm's circular motion obtained by solving equations of motion. The signal-to-noise ratio (SNR) value is inversely correlated with the diffusion coefficient of the center. (**E**)
*Figure 3 continued on next page*

*Figure 3 continued*

Normalized mean step size obtained from experimental data and (F) from results shown in (D). (G) Infrequent rolling results in relocation of the center. (H) Two-phase intermittent search caused by infrequent rolling. (I) Experimental tracking of a sperm exhibiting intermittent search.

The online version of this article includes the following figure supplement(s) for figure 3:

**Figure supplement 1.** Noise in amplitude and frequency of the first and higher harmonics produces no diffusive motion in the circular path's center.

**Figure supplement 2.** The dynamic of thickness of the layer formed by combining the images of the sperm.

**Figure supplement 3.** The speed of the circular path's center in the presence of infrequent rolling.

**Figure supplement 4.** Infrequent rolling increases the area swept by circular motion and thus, the efficiency of surface exploration.

MSD of the circular path's center (*Figure 2E and F*). For any sperm that diffused more rapidly, cell tracking followed by circle fitting was used to characterize the motion of circular path's center (*Figure 3C*).

To identify the flagellar source of the center's diffusion, we solved equations of motion where the amplitude and phase of all harmonics included white Gaussian noises, as described in *Equation 1* (the results are shown in *Figure 3D*). Our simulations suggest that noise in the amplitudes and the phases of first and higher harmonics yielded a noisy $V_P$ without resulting in fluctuations of the center (SI, Section VI). By contrast, the noise in the amplitude of the zeroth harmonic yielded a similar noise in $\Omega$ but not in $V_P$. Therefore, $\Omega$ and $\kappa$ can be expressed as $\Omega(t) = \tilde{\Omega}(1 + \eta_0(t))$ and $\kappa(t) = \tilde{\kappa}(1 + \eta_0(t))$, respectively, such that lower signal-to-noise ratios $\left(\mathrm{SNR} = \frac{\tilde{\kappa}^2}{\langle \kappa^2 \rangle - \tilde{\kappa}^2}\right)$ for identical mean curvatures yielded faster diffusion, whereas higher SNRs yielded more consistent pathways (*Figure 3D*). We measured the normalized mean step size $\langle \delta^2 \kappa^2 \rangle$ and the corresponding SNR and found that our experimental data were consistent with the simulation-based results, except for a constant as shown in *Figure 3E and F*. That is, as suggested by simulations, the normalized mean step size was found to be inversely correlated with the SNRs: $\langle \delta^2 \kappa^2 \rangle \propto \mathrm{SNR}^{-1}$.

As previously published (*Ma et al., 2014*; *Goldstein et al., 2009*), these fluctuations in the circular path's center correspond to non-thermal noise in the sperm flagellar beating, rather than thermally driven fluctuations. Approximating sperm as a rod with length of ~80 μm and diameter of ~5 μm, the rotational diffusion caused by thermal fluctuations is in the other of $10^{-6}\mathrm{s}^{-1}$, which yields a $SNR \sim 10^7$ for $\tilde{\kappa} \sim (5 \times 10^{-3}) \mu m^{-1}$. This order of $SNR$ estimated for thermal fluctuations are far greater than our measured values and the experimental resolution; therefore, the contribution of thermal noise to the sperm diffusive circular motion is negligible.

Diffusive circular motion might be abruptly interrupted by infrequent rollings (*Figure 1I*, *Figure 3G and H*). In this case, the center of the circular path not only diffuses in time but also relocates ballistically upon rolling in a random direction with $V_r$ (*Figure 3H,I*), such that $\frac{D}{V_r} \sim 10^{-6} - 1$ μm (SI, Section VII). This two-phase motion is an intermittent search, with greater efficiency in surface exploration than with normal diffusion (*Bénichou et al., 2011*). This increase in the efficiency of exploration may be interpreted as a decrease in the probability of revisiting previously swept spots. More precisely, the average area swept by the sperm through diffusion is proportional to the square root of diffusion time $\langle A(t) \rangle \propto \sqrt{t}$. Assuming that $T_i$ is the time frame between the $(i-1)$th and $i$th relocations, the areas swept with and without relocation are proportional to $\sum_i \sqrt{T_i}$ and $\sqrt{\sum_i T_i}$, respectively. Because $\sqrt{\sum_i T_i} \leq \sum_i \sqrt{T_i}$, infrequent rolling increases the area swept by the sperm (see SI, Section VII).

## Rolling and wall-dependent navigation

In addition to active swimming and external flow, the surrounding walls contribute to sperm motion (*Elgeti et al., 2010*). For a sperm that is positioned away from the walls (distance from the wall > the sperm length), the wall's effect on the swimmer is known to be a drift velocity that can be either attractive or repulsive depending on the swimmer's angle with respect to the wall (*Elgeti et al., 2010*). The model proposed for studying this drift velocity is based on positing the swimmer as a force dipole that includes the propulsive force provided by the flagellum as well as the corresponding drag force (*Elgeti et al., 2010*). However, this model can be used when the sperm exhibits fully

symmetric flagellation. We therefore developed a new swimmer model that includes the torque caused by asymmetric beating. The proposed model is depicted in *Figure 4—figure supplement 1A*, in which $f$ is the propulsive force, $f''$ is the perpendicular force corresponding to the torque, and $f'$ is the drag force required for the torque-free condition. We then carried out finite element method simulations in a cylindrical domain, like our microfluidic quiescent reservoir, and solved the Stokes and mass conservation equations for our model to find the velocity field imposed by the sperm active swimming.

The velocity field imposed by flagellar beating for $\widetilde{\Delta\theta} = 0 - 15°$ is shown in *Figure 4—figure supplement 1B*. Integrating net flow in the $y$ direction imposed on the sperm body that is caused by no-slip walls, we calculated the drift velocity toward the wall (*Figure 4—figure supplement 1C*). The simulation results indicated that the motion of the microswimmer is influenced to a lesser degree by nearby boundaries as the circular component emerges in the motility (SI, Section VIII). This decay in the drift velocity caused by the circular component of motion was also predicted by the analytical solution derived from a Stokeslet (*Ardekani and Stocker, 2010*; *Li and Ardekani, 2014*) description (SI, Section IX). The attraction of the sperm toward the wall in the presence of circular behavior can be described by the following equation:

$$U_w = U_w^p\left(1 - \frac{3}{2}\sin\left(2\widetilde{\Delta\theta}\right)\right),$$ (11)

where $U_w$ and $U_w^p$ are the drift velocities with and without the circular motion. Calculating the average far-field drift velocity imposed on the sperm during one round of circulation, we found that the average magnitude of the drift velocity in one round (<1 µm/s) is much smaller than random fluctuations ((>540 µm/s) of the circular path's center (*Figure 4—figure supplement 1D* and SI, Section X); therefore, the sperm's circular motion is more strictly controlled by the level of asymmetry in its flagellum and diffusivity than by distant walls.

Frequent rollings, however, negate the contribution of circular components of the motion, producing a drift velocity as if the flagellation is symmetric and the sperm behaves like a dipole swimmer,

$$\tilde{U}_w = U_w^p\left(1 - \frac{3}{2}\widetilde{\Pi(t)}\sin\left(2\widetilde{\Delta\theta}\right)\right) \approx U_w^p.$$ (12)

We reiterate that $\widetilde{\Pi(t)} \ll 1$. *Equation 12* states that frequent rollings make the sperm more readily disposed to physical boundaries at the far

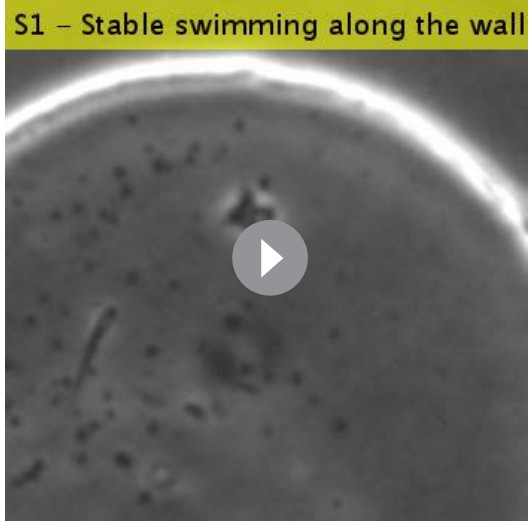

**Video 7.** Rolling sperm motion along the wall ($S_1$).
https://elifesciences.org/articles/68693#video7

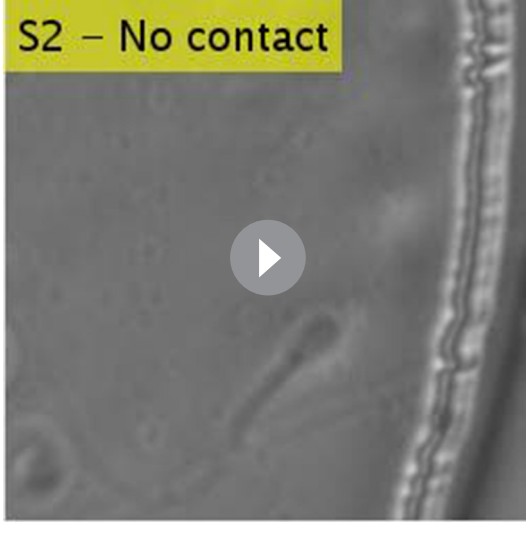

**Video 8.** Non-rolling sperm with high $\Omega_{in}$ does not contact the wall ($S_2$).
https://elifesciences.org/articles/68693#video8

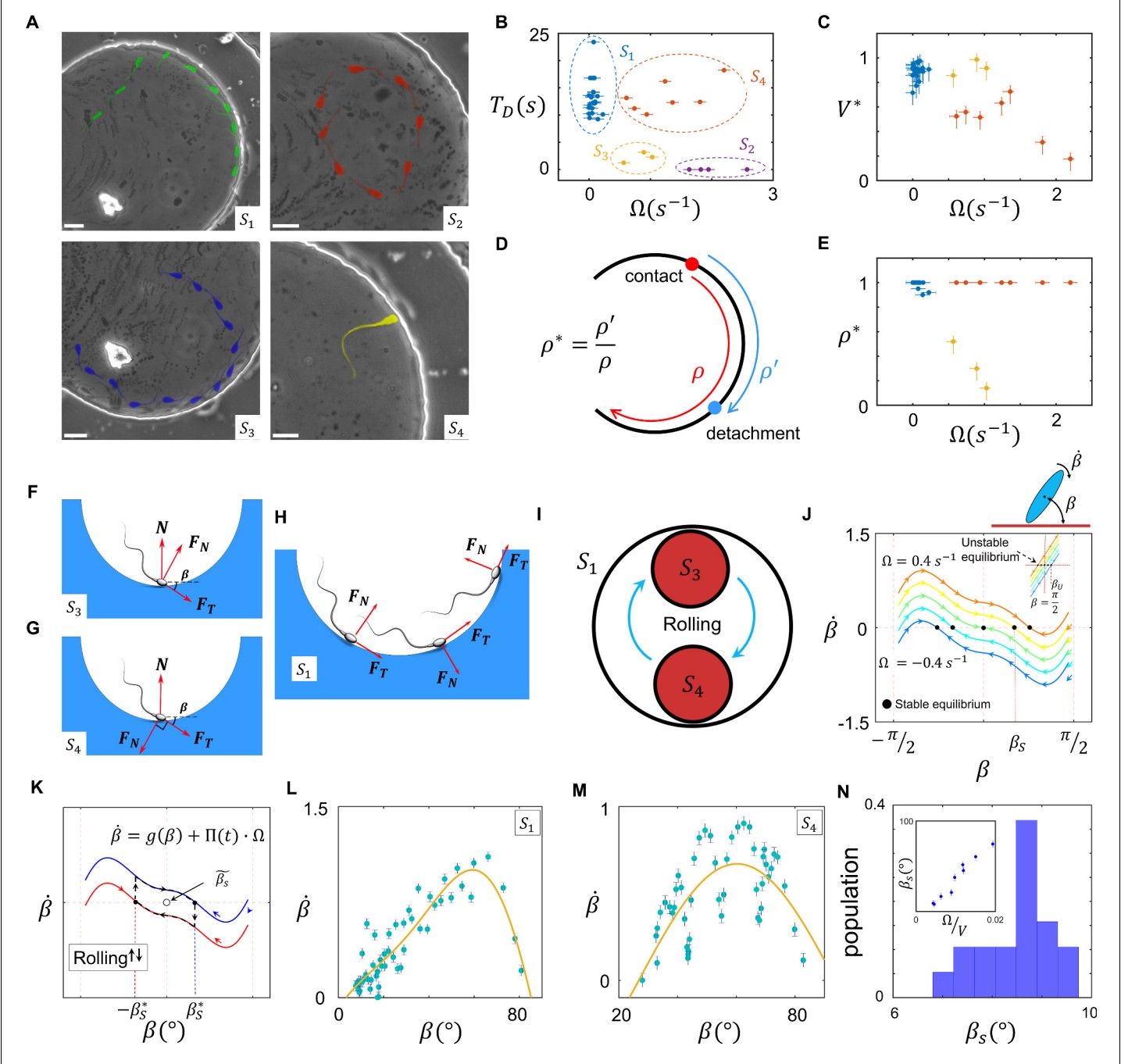

**Figure 4.** Sperm–wall interactions. (**A**) Sperm–wall interaction categories. Rolling sperm rotate and align with the wall upon contact to swim stably along it ($S_1$). Non-rolling sperm exhibiting circular motion either do not contact the wall ($S_2$), detach from the wall after temporarily swimming along it ($S_3$), or swim slowly along the wall ($S_4$) depending on the magnitude and direction of angular velocity. (**B**) Sperm detention time on the wall, that is, $T_D$. (**C**) Sperm normalized velocity on the wall, that is, $V^*$. (**D**) A schematic to illustrate the definition of $\rho^*$. (**E**) $\rho^*$ versus $\Omega$. (**F**) Free body diagram for the $S_3$, (**G**) $S_4$, and (**H**) $S_1$ categories. (**I**) Rolling results in alterations between $S_3$ and $S_4$, forming $S_1$. (**J**) The phase curves describing the dynamics of the angle between the sperm and wall after contact. Filled dots are stable points. Note that negative and positive $\beta_S$ values correspond to the $S_3$ and $S_4$ categories, respectively. (**K**) Rolling can be modeled as a transition between two-phase curves obtained for $\pm\Omega$. A frequent transition between $\pm\Omega$ results in $\tilde{\beta}_S \ll \beta_S^*$. (**L**) The phase curve for a rolling sperm. The final angle between the sperm and wall is ~10°. (**M**) The phase curve for a non-rolling sperm. The final angle between the sperm and wall is ~30°. (**N**) The distribution of $\tilde{\beta}_S$ for 20 rolling sperm. $\tilde{\beta}_S$ is linearly related to $\frac{\Omega}{V}$ for non-rolling sperm (inset plot).

The online version of this article includes the following figure supplement(s) for figure 4:

*Figure 4 continued on next page*

field.

Sperm near-field interactions with its nearby walls fit in one of the four categories shown in *Figure 4A*. A rolling sperm reorients upon wall contact and swims along the wall ($S_1$, *Video 7*), whereas with a non-rolling sperm, depending on $\Omega$ and the location of the circular path's center at the contact point, three other behaviors were observed. At a high magnitude of $\Omega$, the sperm do not contact the wall, maintaining their circular motion ($S_2$, *Video 8*). At lower magnitudes of $\Omega$, the sperm contact the wall and, depending on the location of the path's center relative to the contact point, either swim near the wall temporarily and detach ($S_3$, *Video 9*) or swim more slowly along the wall (compared with $S_1$) with a tilted orientation with respect to the wall ($S_4$, *Video 10*). We compared these four types of sperm–wall interactions quantitatively using the time of sperm detention on the wall ($T_D$, *Figure 4B*), sperm velocity on the wall divided by its velocity before wall contact ($V^*$, *Figure 4C*), and the normalized length of detention $\rho^*$ (*Figure 4D,E*). Based on the measurements shown in *Figure 4B–E*, we found that the $S_1$ and $S_3$ categories yield similar $V^*$ values (close to 1), whereas for $S_4$, $V^*$ is smaller than 1 and for $\Omega > 1.5 \text{ s}^{-1}$ the sperm do not migrate along the wall. Furthermore, $S_1$ and $S_4$ yield similar $\rho^*$ values, close to 1, whereas for $S_3$, $\rho^*$ is much smaller than 1, indicating temporary detention on the wall. A simple yet insightful approach to understanding these four categories involves surface contact force analysis (*Marion, 2013*), where the sperm contact with the wall can be modeled by a positive force that is perpendicular to the surface $N$.

Suppose that a sperm swims along a wall at an angle $\beta$ (*Figure 4F*). Under a zero net force constraint, the normal surface force becomes $N = F_T \sin(\beta) - F_N \cos(\beta)$, where $F_T$ and $F_N$ are the propulsive and perpendicular forces produced by the sperm, respectively. The threshold angle ($\beta_{th}$) that corresponds to the $N = 0$ situation is equal to $\tan^{-1} \gamma$, where $\gamma = \frac{F_N}{F_T}$. For $\beta < \beta_{th}$, $N$ becomes negative and no contact occurs ($S_2$). Because an increase in $\gamma$ leads to higher $\beta_{th}$, sperm with greater $\gamma$ values are less likely to contact and follow the wall. For $\beta$ values greater than $\beta_{th}$, where sperm–wall contact occurs, greater $F_N$ yields a smaller $N$ and leads

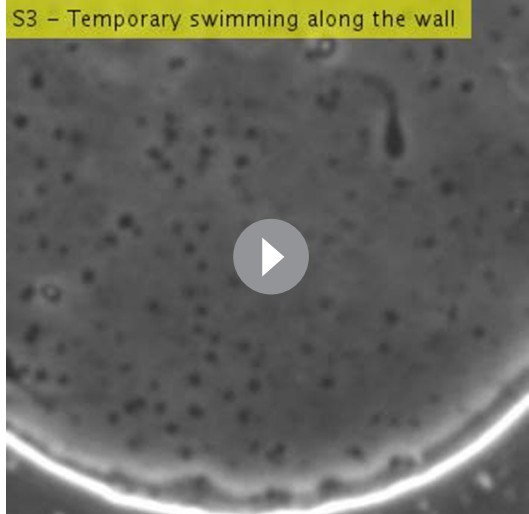

**Video 9.** Non-rolling swim near the wall temporarily and detach ($S_3$).
https://elifesciences.org/articles/68693#video9

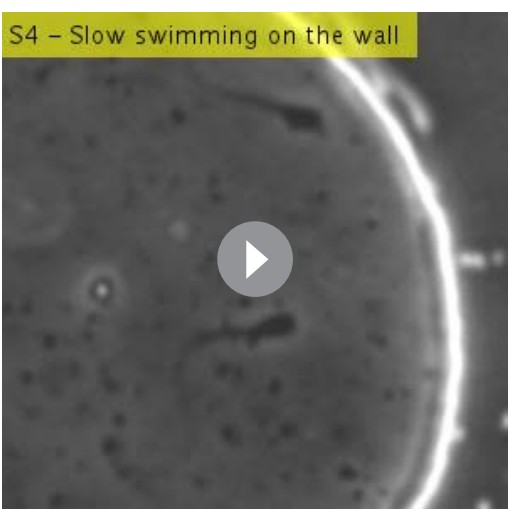

**Video 10.** Non-rolling sperm swim slowly along the wall with a tilted orientation with respect to the wall ($S_4$).
https://elifesciences.org/articles/68693#video10

to easier detachment from the wall ($S_3$). When the direction of the sperm perpendicular force at the contact point is against the wall ($N = F_T \sin(\beta) + F_N \cos(\beta_s)$), a greater surface force is exerted, yielding a stronger attachment to the wall (*Figure 4G*). In this configuration, however, the parallel-to-the-wall components of the perpendicular and propulsive forces thwart each other and cause slower sperm motion along the wall. Therefore, asymmetric flagellation perturbs sperm motion along the wall by decreasing either its detention time ($S_3$) or velocity ($S_4$) on the wall. However, the rolling component functions as a switch between $S_3$ and $S_4$, and not only guarantees longer detention but also increases sperm velocity along the wall (*Figure 4H,I*).

Based on previous theoretical and computational studies on hydrodynamic interactions of a sperm (or a bacteria) with a wall (*Drescher et al., 2011*; *Schaar et al., 2015*; *Elgeti and Gompper, 2013*; *Rode et al., 2019*; *Elgeti and Gompper, 2016*), we developed a simple hydrodynamic model of sperm–wall interaction at the lubrication limit. Solving Stokes and mass conservation equations (SI, Section XI), we plotted the phase curve ($\dot{\beta}$vs$\beta$) of sperm dynamics after contacting the wall (*Figure 4J*). Note that $\dot{\beta}$ is the effect of the wall superimposed with intrinsic $\Omega$. For $\Omega = 0$, stability ($\dot{\beta} = 0$) occurs at $\beta_S = 0$, implying that the final orientation of a symmetrically beating sperm with respect to the wall is 0. For $\Omega > 0$, corresponding to $S_4$, $\beta_S > 0$ is the final angle between the sperm and the wall. For $\Omega < 0$, corresponding to $S_3$, $\beta_S < 0$ and $\frac{d\beta}{dt} > 0$ at $\beta = 0$, suggesting that swimming parallel to the wall is unstable, and the sperm detaches from the wall with $\beta_S$. *Figure 4J* shows that the instability that occurs at $\beta_U = \frac{\pi}{2}$ (for $\Omega = 0$) changes with $\Omega$ as well.

To include the contribution of rolling in sperm dynamics after wall contact, we modeled rollings as transitions between two curves in the phase space with positive and negative values of $\Omega$ (*Figure 4K*). We then can rewrite $\dot{\beta}$ and include $\Pi(t)$ such that:

$$\dot{\beta} = g(\beta) + \Pi(t)\Omega. \tag{13}$$

Note that $g(\beta)$ is the curve in the phase space that corresponds to $\Omega = 0$. Insofar as stability occurs at $\dot{\beta} = 0$,

$$\beta_S^*(t) = g^{-1}(-\Pi(t)\Omega) = \Pi(t)g^{-1}(-\Omega) = \Pi(t)\beta_S, \tag{14}$$

in which $\beta_S^*$ and $\beta_S$ are the stable points with and without taking rolling into account. Because $\widetilde{\Pi(t)} \ll 1$, the average of $\beta_S^*(t)$ is:

$$\widetilde{\beta_S^*} = \widetilde{\Pi(t)}\beta_S \ll \beta_S. \tag{15}$$

*Equation 15* suggests that, at the frequent rolling limit where $\widetilde{\Pi(t)}$ approaches 0, $\widetilde{\beta_S^*}$ approaches 0 as well. Consequently, frequent rollings mitigate the destructive role of $\Omega$ in sperm motion along the wall, thereby yielding faster and longer-lasting swimming along the wall (SI, Section XII). Because at the frequent rolling limit $\widetilde{\beta_S^*}$ is close to 0, we posit that $g\left(\widetilde{\beta_S^*}\right) \approx 0$, and thus $\dot{\beta}$ near the stable point can be written as:

$$\dot{\beta} = g\left(\widetilde{\beta_S^*}\right) + \Pi(t)\Omega \approx \Pi(t)\Omega. \tag{16}$$

Based on *Equation 16*, $\beta(t)$ is a triangular function near the stable point in the form of:

$$\beta(t) \approx \Lambda(t)\Omega + \widetilde{\beta_S^*}. \tag{17}$$

We measured $\dot{\beta}$ experimentally to verify the results obtained from the lubrication theory (*Figure 4J*). The net rotation ($\dot{\beta}$) with respect to the angle between the sperm and wall ($\beta$) that corresponds to $S_1$ and $S_4$ is consistent with the results obtained from our model (*Figure 4J,L,M*). Furthermore, the frequent rollings observed in the $S_1$ category decreased $\widetilde{\beta_S}$ significantly to ~10° (*Figure 4N*) in comparison with $\widetilde{\beta_S}$ for $S_4$, which was greater and linearly correlated with $\frac{\Omega}{V}$ (*Figure 4N*, inset plot). This measurement supports our prediction that rolling enables the sperm to swim stably along the wall by decreasing $\widetilde{\beta_s}$. Measuring $\beta(t)$ during stable swimming along the wall

for 25 sperm that belong to the $S_1$ category, we found that $\beta(t)$ is indeed a triangular function near the stable point, as predicted by *Equation 17* (SI, Section XII).

A unified picture of sperm motion within the quiescent reservoir can be obtained by developing a state diagram and identifying the transitions between the $S_{1-4}$ states, possibly through the diffusivity of the circular motion and rolling. Suppose that, within the quiescent reservoir (radius of $R$), the non-rolling sperm swims in circle (with a radius of $R'$), such that the distance between the centers of the two circles is $d(t)$, which evolves through diffusion (*Figure 5A*). Defining $s(t) = \frac{1}{2}\left(R^2 + R'^2 - d(t)^2\right)$, the circular path does not intersect the reservoir for $s(t) > RR'$ (category $S_2$), whereas for $0 < s(t) \leq RR'$ the angle between the two circles at the intersection point (sperm incidence angle) is greater than $\beta_u$ (category $S_3$); for $s < 0$, the angle at the intersection point is less than $\beta_u$ (category $S_4$). Assuming that at $t = 0$ and the sperm is in the $S_2$ state, the circular path starts to diffuse over time, which leads to transitions between the $S_2$ and $S_3$ states. The average time for the first occurrence of a transition is $\langle T \rangle = \frac{(R-R')^2}{2D}$ (*Figure 5B*). One might expect the diffusion process to continue until $s(t)$ becomes negative and the transition to $S_4$ occurs (*Figure 5B*). However, the angle at which the sperm detaches from the wall after the first contact ($\beta_s$) depends on $\Omega$ rather than the location and angle at which contact occurs (*Figure 5C*). Therefore, after first contact, the sperm returns to the wall at an incidence angle that is identical to the angle at which it detaches from the wall $\beta_s$. Therefore, the transition from $S_3$ to $S_4$ occurs if $\beta_s > \beta_u$. Given that $\beta_s \approx \frac{\Omega}{|g'(0)|}$, $\beta_u \approx \frac{\pi}{2} + \frac{\Omega}{g'\left(\frac{\pi}{2}\right)}$, and $|g'(0)| \ll g'\left(\frac{\pi}{2}\right)$, the condition of such a transition reduces to:

$$\Omega > \frac{\pi}{2}\frac{\left|g'(0)\right|g'\left(\frac{\pi}{2}\right)}{g'\left(\frac{\pi}{2}\right) - \left|g'(0)\right|} \approx \frac{\pi}{2}\left|g'(0)\right|. \tag{18}$$

The prime indicates derivative with respect to $\beta$. The corresponding curvature that satisfies *Equation 18* is $\kappa \sim 0.1\ \mu m^{-1}$, which is much greater than the experimentally observed values. The state diagram of non-rolling sperm–wall interactions and the possible transitions are summarized in *Figure 5D*.

For rolling sperm, the state diagram shown in *Figure 5D* alters significantly. First, infrequent rollings result in abrupt changes in $d(t)$ and thus $s(t)$, causing reversible transitions between $S_2$, $S_3$, and $S_4$ (*Figure 5F*). Second, at the frequent rollings limit, one may write $s(t) \approx \Pi(t)|s|$, so that $\widetilde{s(t)}$ approaches 0 (*Figure 5E*). Consequently, in the presence of frequent rollings, all states transition to $S_1$, where the sperm swims stably along the wall (*Figure 5F*).

## Discussion

In summary, rolling is a component of mammalian sperm motility that is sensitive to ambient viscosity and viscoelasticity. In a solution with low viscosity and viscoelasticity, most sperm exhibit a rolling and progressive motion that is susceptible to external fluid flow (rheotaxis) and rigid physical boundaries (wall-dependent navigation). As ambient viscosity or viscoelasticity increases, the rolling component becomes suppressed and subsequently the sperm swim in diffusive circular paths (surface exploration), a type of motion that is less susceptible to being influenced by external fluid flow or nearby walls. Suppression of rolling was found to be reversible, as sperm migrate into medium with low viscosity or viscoelasticity, rolling reactivates, and thus sperm swimming transitions from surface exploration to progressive motion. Furthermore, we demonstrated that the suppression of rolling in sperm with lower asymmetry in their flagellar beating pattern occurs at higher viscosity or viscoelasticity. Therefore, the suppression of rolling, and thus the onset of surface exploration, depends on both ambient rheological properties and the level of asymmetry in sperm flagellation.

Our results evidenced sperm flagellar beating is intrinsically asymmetric, but frequent rolling counteracts this asymmetric flagellation by alternating the direction of asymmetry, and results in a progressive motion. This progressive motion resulted from frequent rollings found to be key to the sperm rheotaxis and wall-dependent navigations. But why is susceptibility to external fluid flow and nearby walls under dynamic conditions not controlled solely by the level of asymmetry in flagellation so that maximum susceptibility appears with the fully symmetric beating pattern?

We argue that while fully symmetric beating pattern yields an efficient wall-dependent navigation, it does not result in an efficient rheotaxis, because the tilted orientation of the sperm (caused by

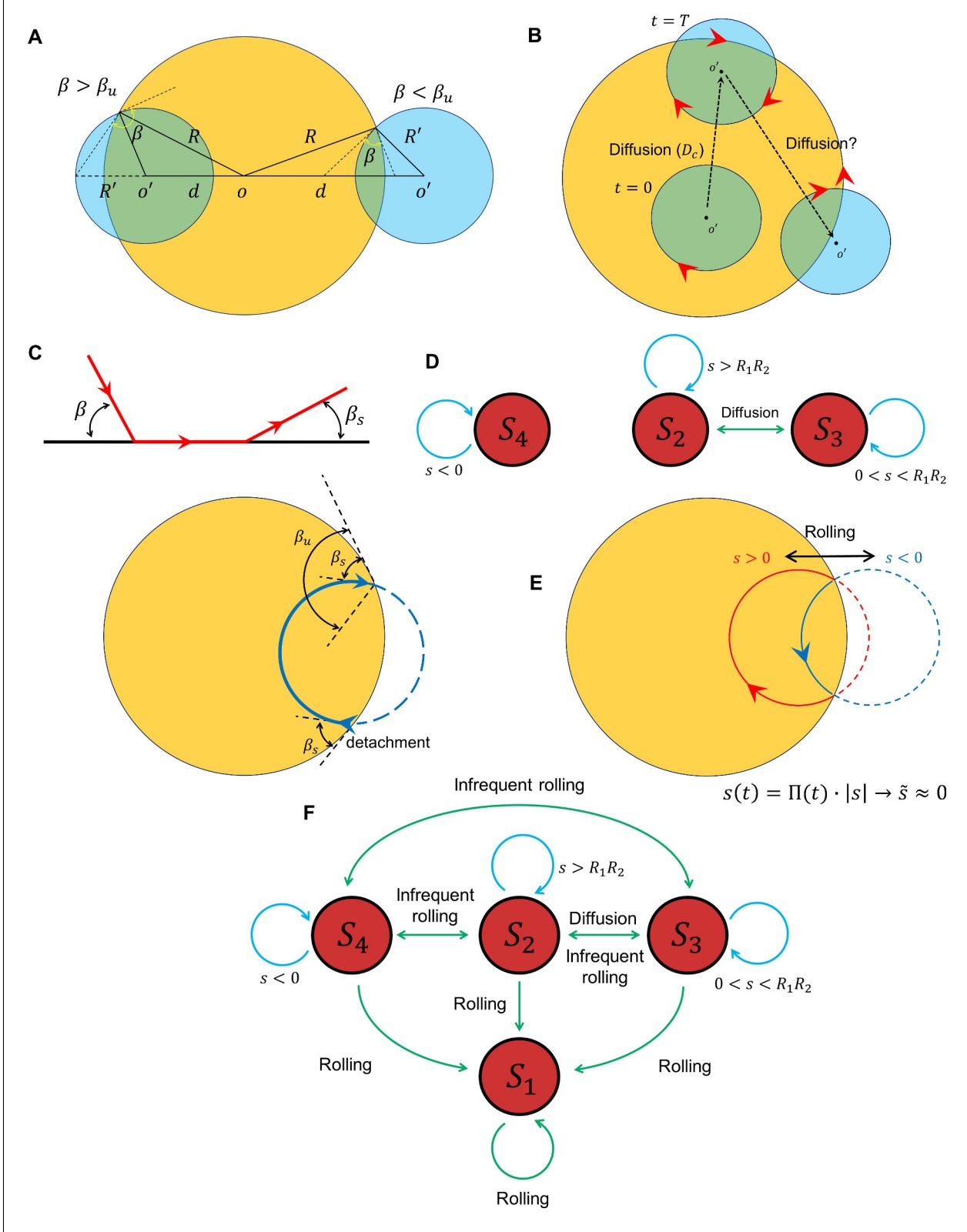

**Figure 5.** Transition between surface exploration and wall-dependent navigation. (**A**) The circular path does not intersect with the reservoir for $s>RR'$, whereas $0<s<RR'$ corresponds to $S_3$ and $s<0$ corresponds to $S_4$. (**B**) Diffusivity in circular motion results in the evolution of $s(t)$, after which $\langle T \rangle S_2$ transforms into $S_3$. (**C**) The angle at which the sperm detaches from the wall $\beta_S$) is independent of the incidence angle at the contact point. After the first contact and detachment, the sperm returns to the wall with $\beta_S$. (**D**) State diagram of non-rolling sperm–wall interactions. (**E**) At the frequent rollings
*Figure 5 continued on next page*

*Figure 5 continued*

limit, the time average of $s(t)$ approaches 0, where $\bar{s} \approx 0$, which corresponds to $S_1$. **(F)** Frequent rollings convert all the states into the $S_1$ state, whereas infrequent rollings result in reversible transitions between $S_2, S_3,$ and $S_4$, as needed for an efficient surface exploration.

rolling) with respect to the surface is needed for efficient rheotaxis, as demonstrated previously (*Kantsler et al., 2014*). More importantly, it is possible that, unlike the asymmetric beating pattern that is modulated by chemical factors (*Suarez, 2008*; *Quill et al., 2003*; *Ramírez-Gómez et al., 2020*), rolling depends on the ambient viscosity and viscoelasticity of the medium alone (*Schiffer et al., 2020*). Because the viscosity and viscoelasticity of the fluid within the female reproductive tract varies across functional regions, the tract possibly regulates sperm navigation by independently controlling: (1) the rolling component through regulating the rheology of the environment and (2) the asymmetry level in the flagellar beating pattern through secreting chemical factors. To show how regulating of asymmetrical beating caused by chemical factors works together with rolling, which is impacted by ambient viscosity and viscoelasticity, more studies regulating both aspects are needed.

Our results also demonstrate that the transition found between fast progressive and slow diffusive circular motions is reversible and occurs through suppression or reactivation of rolling. This finding, in particular, suggests that sperm motion during migration within the female reproductive tract is possibly bimodal. The fast progressive mode is an appropriate swimming behavior for sperm to migrate long distances between different functional regions within the female reproductive tract. This mode of motion may be regulated by the tract through rheotaxis and wall-dependent navigation. Whereas the slow diffusive motion is an appropriate swimming behavior for exploring the functional regions within the female reproductive tract to possibly receive physiological signals from these regions that are essential for the fertilization process. These physiological signals may include specific ligands secreted by the tract (*Chang and Suarez, 2010*), or pH of the functional region (*Marquez and Suarez, 2007*).

Our findings also suggest that elastic properties of the swimming media are key to the suppression of rolling and thus sperm motion. Characterizing the rheological properties of our viscous (4% PVP) and viscoelastic (1% PAM) solutions, we noticed that viscosity of 1% PAM is two orders of magnitude greater than that of 4% PVP solution. Rolling suppression in 1% PAM occurred without a loss in sperm propulsive velocity, whereas rolling suppression in the 4% PVP occurred with a significant loss in the propulsive velocity. Since the storage modulus of 1% PAM solution was two orders of magnitude greater than that of 4% PVP, we, therefore, conclude that the absence of a loss in the sperm propulsive velocity was due to the elastic properties of the solution. Accordingly, the elasticity of the swimming media is a key contributing factor to the sperm navigation within the female reproductive tract.

Our findings revealed the potential role of sperm rolling in mammalian fertilization, which results in a holistic and fundamental understanding of the fertilization process. Furthermore, these results are useful for more practical purposes such as designing technologies to improve fertility in cattle industry, as well as diagnosing and treating human male infertility. Furthermore, our results can be used to design new types of synthetic microswimmers that are responsive to dynamic physical environments, and thus more sensitive while exploring confined spaces.

## Materials and methods

### Sperm sample and culture media preparation

Commercially available cryopreserved bovine semen samples taken from two mature black and white Holstein bulls (5.5 and 6 years of age) were kindly donated by Genex Cooperative (Ithaca, NY) in milk and egg yolk–based extender in plastic straws. The ejaculate concentration was 2.9 and 3 billion cells/mL, respectively, and had a pre-freeze motility of 65%. The semen was thawed at 37°C in a water bath and diluted in a 1:4 ratio with TALP. After dilution, the viscosity of the samples was ~5 mPas. The initial sperm concentration in the thawed semen samples was ~200 million/mL, which was diluted to ~40 million/mL with TALP. The motility of the semen sample after dilution decreased to 20–30%. We used 10 separate semen samples in both the milk and egg yolk–based extender. At

least three replicates were performed for each experiment to validate the accuracy of data and obtain a valid value for error bars.

TALP was prepared as follows: NaCl (110 mM), KCl (2.68 mM), $NaH_2PO_4$ (0.36 mM), $NaHCO_3$ (25 mM), $MgCl_2$ (0.49 mM), $CaCl_2$ (2.4 mM), HEPES buffer (25 mM), glucose (5.56 mM), pyruvic acid (1.0 mM), penicillin G (0.006% or 3 mg/500 mL), and bovine serum albumin (20 mg/mL). To tether the sperm head to the glass surface, we reduced the concentration of bovine serum albumin to 5 mg/mL. To increase the viscosity and viscoelasticity of TALP, we added 1–4% of PVP (weight percent) and 0.25–1% PAM (weight percent).

## Rheological measurements

Dynamic rheological measurements were performed with a rheometer (MCR 501, Anton Paar, Stuttgart, Germany) with a 50 mm parallel plate at a gap of 0.5 mm. The amplitude sweep was conducted from 0.01% to 100% strain with a 1 Hz angular frequency to identify the linear viscoelastic region. The frequency sweep was performed from 1 to 100 $s^{-1}$ (*Tung et al., 2017*) with a constant 1% strain (within the linear viscoelastic region). The viscosity was measured using the steady shear mode with the shear rate from 0.01 to 100 $s^{-1}$. Samples were characterized at 37°C.

## Microfabrication and semen injection

The microfluidic device was made of polydimethylsiloxane using a standard soft lithography protocol. The diameter of the circular quiescent zone was 500 μm and the height of the chamber was 25 μm. Diluted semen was injected into the microfluidic device using gravity and the flow generated in the channel was controlled by changing the height of the semen container. Because sperm rheotaxis occurs under a very low shear rate (0.6 $s^{-1}$), using gravity instead of conventional syringe pumps is more efficient for obtaining and controlling low flow rates.

## Rheotaxis-based sperm isolation and phase-contrast microscopy

To isolate motile bovine sperm inside the quiescent reservoir, we used a microfluidic corral system that isolated motile swimmers based on their ability to move upstream. As we injected the sample at an injection rate of 1.2 $μLh^{-1}$, sperm with motilities higher than 53.2 $μms^{-1}$ could swim upstream and enter the quiescent zone, which was filled with TALP, allowing us to study sperm movement with minimal fluid mechanical noise. Sperm movement was observed with a Nikon Eclipse TE300 inverted phase-contrast microscope (20× and 40× magnifications) and recorded with an Andor Zyla 5.5 sCMOS camera (25 and 50 frames/s).

## Cell tracking and zeroth harmonic measurement

Sperm trajectories and other motility-related characteristics were analyzed using ImageJ and MATLAB. To identify the harmonics of midpiece bending, we first removed noise and background using Gaussian filter and image subtraction. We then binarized the images taken from the sperm at 25 frames/s and measured the deviation of the midpiece (i.e., the segment located at 10 ± 1 μm from the head) from the centerline using the optical flow Flareback method. Taking the fast Fourier transform of the bending, we identified the zeroth, first, and second harmonics of the bending signal. A simple method for measuring the amplitude of the zeroth harmonic involves measuring the maximum bending toward the left ($y_L$) and right ($y_R$) sides of the swimmer. Therefore, the magnitude of the zeroth harmonic can be calculated using the following equation:

$$a_0 = \frac{|y_L - y_R|}{2}. \tag{19}$$

The main advantage of our method is its simplicity, as tracking the whole flagellum was not required for measurement of zeroth harmonic.

## Numerical simulation

### Beating pattern

To model the beating pattern of a sperm, we posited that the flagellation in one beat could be described by a sine wave at a temporal interval of $[\pi - \phi_0, \phi_0]$, such that $\phi_0 \in [\pi, 2\pi]$. $\phi_0$ determines the level of asymmetry in the beating pattern. Thereafter, by evenly extending the function, we

obtained the beating patterns, where $\phi_0$ determined the asymmetry in the beating. For example, $\phi_0 = 2\pi$ corresponds to $\sigma = 1$, and thus symmetric beating, whereas $\phi_0 < 2\pi$ results in $\sigma < 1$, and thus asymmetric beating. We then applied a fast Fourier transform on the beating patterns to determine their temporal frequencies. These steps were performed using MATLAB (version R2017a).

## Finite element method simulations

To obtain the velocity field imposed by the swimmer model shown in *Figure 4—figure supplement 1* and determine the far-field hydrodynamic interactions, we first imported the cylindrical structure of the quiescent zone to the COMSOL MULTIPHYSICS (version 5.2) platform. Two orthogonal Gaussian pulse functions (defined in the $x$ and $y$ directions) were used to define each point force in the swimmer model. The mathematical form of the pulse is a 2D Gaussian distribution, as follows:

$$f\delta(x-x_0)\delta(y-y_0) = \frac{f}{2\pi\sigma_x\sigma_y}e^{\frac{-(x-x_0)^2}{2\sigma_x^2}}e^{\frac{-(y-y_0)^2}{2\sigma_y^2}}. \tag{20}$$

We used $x_0, y_0$ to move and $\sigma_x$, $\sigma_y$ to focus the point forces arbitrarily. This strategy was chosen to lower the computational cost and avoid issues related to using small volumetric forces and their associated meshing problems in the finite element method. Finally, assuming that sperm swim in a quasi-2D plane that is located 5 µm below and parallel to the top surface, we solved the Stokes (*Equation 21*) and mass conservation equations for varying $\Delta\theta$ values:

$$\nabla p - \mu\nabla^2 v = \sum f_i\delta(r-r_i). \tag{21}$$

In *Equation 21*, $p$ is the pressure, $\mu$ is the dynamic viscosity of the TALP medium ($3.2\,\mathrm{mPas}$), $v$ is the fluid velocity, $r$ is the position, $r_i$ is the position of the point force $f_i$, and $\delta$ is the Dirac delta function. The results obtained from this section are demonstrated in *Figure 4—figure supplement 1B*. Then, by integrating the velocity field imposed by the sperm, we obtained the drift velocity toward the wall, as shown in *Figure 4—figure supplement 1C*.

To find the torque imposed on the sperm in near-field conditions through lubrication approximation, we used a finite element method to solve Stokes and mass conservation equations for the configuration shown in Appendix 1. Given that the contribution of pressure in the stress tensor dominates that of viscous stress (SI, Section XI) ($pI \gg \mu(\nabla v + \nabla v^T)$), we extracted the pressure exerted on the sperm (*Figure 4—figure supplement 5*) at varying incident angles ($-90° < \beta < 90°$) for a constant progressive velocity ($V_p = 80\,\mu\mathrm{m/s}$). The torque exerted on the sperm from the wall and the corresponding angular velocity was then calculated:

$$\dot{\beta} = \frac{d\beta}{dt} = \frac{\alpha}{\xi_N}\frac{\int_0^L p(x-x_{cm})dx}{\int_0^L (x-x_{cm})^2 dx}. \tag{22}$$

In *Equation 22*, $p$ is pressure, $x_{cm}$ is the coordinate of the sperm center of mass, $L$ is the sperm length, and $\alpha$ is a fitting parameter. Note that $\dot{\beta}$ is linearly correlated to $V_p$.

## Acknowledgements

The authors would like to thank SS Suarez for fruitful discussions about the biological aspects of the work. We also thank DL Koch, CK Tung, J Fan, and M Esmaily for helpful discussions about hydrodynamic interactions, the dipole swimmer model, and the surface force analysis used here. The authors also thank our dearest KJ Donaghy for helping us edit this manuscript. This work was performed in part at the Cornell NanoScale Facility, a member of the National Nanotechnology Coordinated Infrastructure (NNCI), which is supported by the National Science Foundation (Grant NNCI-2025233).

## Additional information

### Funding

| Funder | Grant reference number | Author |
|---|---|---|
| Cornell University | | Alireza Abbaspourrad |
| National Science Foundation | I20252 | Alireza Abbaspourrad |

The funders had no role in study design, data collection and interpretation, or the decision to submit the work for publication.

### Author contributions

Meisam Zaferani, Conceptualization, Data curation, Software, Formal analysis, Validation, Investigation, Visualization, Methodology, Writing - original draft, Writing - review and editing; Farhad Javi, Validation, Methodology, Writing - review and editing; Amir Mokhtare, Software, Validation, Writing - review and editing; Peilong Li, Formal analysis, Methodology; Alireza Abbaspourrad, Conceptualization, Supervision, Funding acquisition, Validation, Writing - review and editing

### Author ORCIDs

Alireza Abbaspourrad (iD) https://orcid.org/0000-0001-5617-9220

### Decision letter and Author response

Decision letter https://doi.org/10.7554/eLife.68693.sa1
Author response https://doi.org/10.7554/eLife.68693.sa2

## Additional files

### Supplementary files

• Transparent reporting form

### Data availability

All data needed to evaluate the conclusions in the paper are present in the paper and/or the Supplementary Materials. All data related to this paper are deposited in https://doi.org/10.5061/dryad.ngf1vhhtd.

The following dataset was generated:

| Author(s) | Year | Dataset title | Dataset URL | Database and Identifier |
|---|---|---|---|---|
| Zaferani M, Javi F, Mokhtare A, Li P, Abbaspourrad A | 2021 | Data from: Rolling controls sperm navigation in response to the dynamic rheological properties of the environment | https://doi.org/10.5061/dryad.ngf1vhhtd | Dryad Digital Repository, 10.5061/dryad.ngf1vhhtd |

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

# Appendix 1

## I Rheology of the sperm media

To increase the viscosity and viscoelasticity of the standard TALP medium, we added 1–4% of PVP and 0.25–1% of PAM. The rheological properties of the prepared solutions are shown in *Figure 1— figure supplement 1*. Note that $G'$ of PVP-based solutions with percentages lower than 4% was very low and not detected by the rheometer.

## II Modeling the sperm beating patterns and Fourier analysis

To model beating patterns that resemble those of sperm flagella, we studied the pattern in one cycle of flagellar beating using a traveling sine wave with a temporal phase of $\phi(t) = \omega t$ (with $\omega = 40$ Hz) in the range of $\pi - \phi_0 \leq \phi(t) \leq \phi_0$, such that $\phi_0 \in [\pi, 2\pi]$. Notably, $\phi_0 = 2\pi$ corresponds to a fully symmetric beating pattern and thus, $\sigma = 1$, whereas lower values of $\phi_0$ correspond to lower $\sigma$ values and higher asymmetry in the beating pattern.

Positing that the beating pattern will identically repeat in time, we constructed the even extension of the partial sine wave to form the flagellar beating function over time. *Figure 2—figure supplement 1* shows the modeled beating patterns of untethered sperm, which resemble the flagellar beating observed inside the quiescent zone, especially that of the mid- and principal piece, which we are interested in. To model the beating pattern, we considered the length preservation constraint as well because the sine wave form does not preserve the length in one beat.

### Length preservation constraint

Knowing that the sperm length ($L_0$) is constant over time, we must find the position of its tip on the x axis in time ($L(t)$), to satisfy the length preservation constraint:

$$\int_0^{L(t)} \left(1 + y'^2\right)^{\frac{1}{2}} dx = L_0. \tag{S1}$$

Finding $L(t)$ enables us to define the beating pattern in $x \in [0, L(t)]$ and thereby solve the problem of length preservation.

Assuming that $\forall i, (ka_i)^2 \ll 1$:

$$\left(1 + y'^2\right)^{\frac{1}{2}} \sim 1 + \frac{1}{2}y'^2 = 1 + \frac{1}{2}k^2(a_0 \sin kx - a_1 \sin(\omega t - kx))^2. \tag{S2}$$

Plugging *Equation S2* into *Equation S1* yields *Equation S3*:

$$\int_0^{L(t)} \left\{1 + \frac{1}{2}k^2(a_0 \sin kx - a_1 \sin(\omega t - kx))^2\right\} dx = L_0. \tag{S3}$$

The following calculations simplify *Equation S3* and break it down into $I_1$ and $I_2$:

$$\int_0^{L(t)} \left\{1 + \frac{1}{2}k^2 a_0^2 \sin^2 kx + \frac{1}{2}k^2 a_1^2 \sin^2(\omega t - kx) - k^2 a_0 a_1 \sin kx \sin(\omega t - kx)\right\} dx = L_0$$

$$\int_0^{L(t)} \left\{\begin{array}{l} 1 + \frac{1}{2}k^2 a_0^2 \left(\frac{1 - \cos 2kx}{2}\right) + \frac{1}{2}k^2 a_1^2 \left(\frac{1 - \cos 2(\omega t - kx)}{2}\right) \\ - k^2 a_0 a_1 \left(\frac{\cos(2kx - \omega t) - \cos \omega t}{2}\right) \end{array}\right\} dx = L_0. \tag{S4}$$

$$\int_0^{L(t)} \left\{1 + \frac{1}{4}k^2 a_0^2 + \frac{1}{4}k^2 a_1^2 + \frac{1}{2}k^2 a_0 a_1 \cos \omega t\right\} dx$$

$$- \int_0^{L(t)} \left\{\frac{1}{4}k^2 a_0^2 \cos 2kx + \frac{1}{4}k^2 a_1^2 \cos(2\omega t - 2kx) + \frac{1}{2}k^2 a_0 a_1 \cos(2kx - \omega t)\right\} dx = L_0. \tag{S5}$$

$$I_1 - I_2 = L_0. \tag{S6}$$

$I_1$ has no $x$-dependent variable and can be calculated as follows:

$$I_1 = \int_0^{L(t)} \left\{ 1 + \frac{1}{4}k^2 a_0^2 + \frac{1}{4}k^2 a_1^2 + \frac{1}{2}k^2 a_0 a_1 \cos \omega t \right\} dx$$

$$= L(t) \left\{ 1 + \frac{1}{4}k^2 a_0^2 + \frac{1}{4}k^2 a_1^2 + \frac{1}{2}k^2 a_0 a_1 \cos \omega t \right\}. \tag{S7}$$

To calculate $I_2$, we first calculate the first term of the equation:

$$\int_0^{L(t)} \frac{1}{4}k^2 a_0^2 \cos 2kx \, dx = \frac{1}{8}ka_0^2 \sin(2kL(t)) = \frac{1}{8}ka_0^2 \sin\left(\frac{4\pi L(t)}{L}\right)$$

$$= \frac{1}{8}ka_0^2 \sin\left(4\pi \frac{L(t) - L}{L}\right). \tag{S8}$$

Knowing that

$$L(t) - L = \delta(t) \ll L, \tag{S9}$$

we can use the following approximation:

$$\frac{1}{8}ka_0^2 \sin\left(4\pi \frac{L(t) - L}{L}\right) \approx \frac{\pi}{2}ka_0^2 \frac{\delta(t)}{L} = \frac{1}{4}k^2 a_0^2 \delta(t). \tag{S10}$$

Using the same method to calculate the second and third terms of $I_2$ shows that:

$$I_2 \propto k^2 \left(a_0^2 + a_0^2 + 2a_0 a_1\right)\delta(t). \tag{S11}$$

Given that $\delta(t) \ll L$, and $L(t)$, $I_2$ can be neglected compared to $I_1$.
Therefore, we may write $L(t)$ as follows:

$$L(t) = L\left(1 + \frac{1}{4}k^2 a_0^2 + \frac{1}{4}k^2 a_1^2 + \frac{1}{2}k^2 a_0 a_1 \cos \omega t\right)^{-1}$$

$$\approx L\left(1 - \frac{1}{4}k^2 a_0^2 - \frac{1}{4}k^2 a_1^2 - \frac{1}{2}k^2 a_0 a_1 \cos \omega t\right). \tag{S12}$$

After modeling the flagellar beating pattern and considering length preservation and small amplitude constraints, we analyzed the beating pattern and the resulting sperm motions by applying the Fourier transform to identify temporal frequencies (*Figure 2—figure supplement 2*). Interestingly, with increasing temporal asymmetry in the beating pattern and thus $\sigma$, the frequency of the main (first) harmonic increases, whereas its amplitude decreases. Moreover, the zeroth and second harmonics simultaneously appear in the frequency domain; however, the amplitudes of the second and higher harmonics are much smaller than that of the zeroth and first harmonics, which enables us to approximate the level of asymmetry as follows:

$$\sigma \approx \frac{a_1 - a_0}{a_1}. \tag{S13}$$

## III Resistive force theory

In this section, we used resistive force theory to derive equations describing the forces produced by each segment of the flagellum and follow the presentation used by Friedrich et al. and Saggiorato et al. The velocity of each segment in the $y$ direction ($V$) can be decomposed into its tangential and normal components $V_T$ and $V_N$, respectively, using $\alpha$, which is the tangential angle (*Equations S14– S17*):

$$V = \frac{\partial y}{\partial t} \tag{S14}$$

$$V_T = V \sin \alpha \tag{S15}$$

$$V_N = V \cos \alpha \tag{S16}$$

$$\tan \alpha = \frac{\partial y}{\partial x}. \tag{S17}$$

According to the resistive force theory, the forces produced by each element in the tangential and normal direction are linearly related to the velocity in those directions:

$$f_T = -\xi_T V_T \tag{S18}$$

$$f_N = -\xi_N V_N. \tag{S19}$$

where $\xi_T$ and $\xi_N$ are drag coefficients in the tangential and normal direction, respectively. Because the amplitudes of all harmonics are small in comparison to sperm length, we may posit the following assumptions:

$$\forall n, a_n \ll L \rightarrow \tan \alpha \approx \sin \alpha \approx \alpha \tag{S20}$$

$$\cos \alpha \approx \left( 1 - \frac{\alpha^2}{2} \right). \tag{S21}$$

Using the approximations in *Equations S20 and S21*, we can write the tangential and normal velocities and forces using *Equations S22–S25*:

$$V_T \approx \left( \frac{\partial y}{\partial t} \right) \left( \frac{\partial y}{\partial x} \right) \tag{S22}$$

$$V_N \approx \left( \frac{\partial y}{\partial t} \right) \left( 1 - \frac{1}{2} \left( \frac{\partial y}{\partial x} \right)^2 \right) \tag{S23}$$

$$f_T = -\xi_T \left( \frac{\partial y}{\partial t} \right) \left( \frac{\partial y}{\partial x} \right) \tag{S24}$$

$$f_N = -\xi_N \left( \frac{\partial y}{\partial t} \right) \left( 1 - \frac{1}{2} \left( \frac{\partial y}{\partial x} \right)^2 \right). \tag{S25}$$

The force produced by each segment in the *x* and *y* directions can be described as follows:

$$f_x = f_T \cos \alpha - f_N \sin \alpha \tag{S26}$$

$$f_y = f_T \sin \alpha + f_N \cos \alpha. \tag{S27}$$

Plugging *Equations S24 and S25* into *Equations S26 and S27*:

$$f_x = -\xi_T \left( \frac{\partial y}{\partial t} \right) \left( \frac{\partial y}{\partial x} \right) \left( 1 - \frac{1}{2} \left( \frac{\partial y}{\partial x} \right)^2 \right) + \xi_N \left( \frac{\partial y}{\partial t} \right) \left( \frac{\partial y}{\partial x} \right) \left( 1 - \frac{1}{2} \left( \frac{\partial y}{\partial x} \right)^2 \right) \tag{S28}$$

$$f_y = -\xi_T \left(\frac{\partial y}{\partial t}\right)\left(\frac{\partial y}{\partial x}\right)^2 \pm \xi_N \left(\frac{\partial y}{\partial t}\right)\left(1 - \frac{1}{2}\left(\frac{\partial y}{\partial x}\right)^2\right)\left(1 - \frac{1}{2}\left(\frac{\partial y}{\partial x}\right)^2\right), \tag{S29}$$

yields *Equations S30 and S31*, which describe the forces produced by the flagellum in the $x$ and $y$ directions.

$$f_x = -(\xi_N - \xi_T)\left(\frac{\partial y}{\partial t}\right)\left(\frac{\partial y}{\partial x}\right) \tag{S30}$$

$$f_y = -\xi_N \left(\frac{\partial y}{\partial t}\right) + (\xi_N - \xi_T)\left(\frac{\partial y}{\partial t}\right)\left(\frac{\partial y}{\partial x}\right)^2. \tag{S31}$$

## IV Force and torque produced by the flagellum

Force in the $x$ direction

The forces produced by a segment of the flagellum moving with $y(x,t)$ in the tangential and normal directions can be described by *Equations S31 and S32*, where $\xi_T$ and $\xi_N$ are the corresponding drag coefficients, respectively. Plugging *Equation S32* into *Equation S30* yields *Equation S34*.

$$y(x,t) = \sum_{n=0} a_n \cos(n\omega t - kx) \tag{S32}$$

$$f_x = (\xi_N - \xi_T)\left(\sum_{n=0} a_n(n\omega)\sin(n\omega t - kx)\right)\left(\sum_{n=0} -a_n k \sin(n\omega t - kx)\right) \tag{S33}$$

$$f_x = -k\omega(\xi_N - \xi_T)\left(\sum_{i,j} i a_i a_j \sin(i\omega t - kx)\sin(j\omega t - kx)\right). \tag{S34}$$

For $i \neq j$, the time average of $f_x$ would be zero, thus we only retain terms with $i = j$. This yields *Equations S35 and S36*.

$$f_x = -k\omega(\xi_N - \xi_T)\left(\sum_{n=1} n a_n^2 \sin^2(i\omega t - kx)\right) \tag{S35}$$

$$f_x = -\frac{1}{2}k\omega(\xi_N - \xi_T)\left(\sum_{n=1} n a_n^2 (1 - \cos 2(i\omega t - kx))\right). \tag{S36}$$

By calculating the time average of *Equation S36*, the average force produced by each segment of the flagellum in the $x$ direction can be described by *Equation S37*.

$$\tilde{f}_x = \frac{1}{T}\int_0^T f_x dt = -\frac{1}{2}k\omega(\xi_N - \xi_T)\sum_{n=1} n \tilde{a}_n^2. \tag{S37}$$

Integrating $\tilde{f}_x$ over the flagellum, the total force and velocity produced in the $x$ direction are:

$$\tilde{F}_x = L\tilde{f}_x = -\frac{1}{2}k\omega L(\xi_N - \xi_T)\sum_{n=1} n \tilde{a}_n^2 = -\xi_T \tilde{V}_p L \rightarrow \tilde{V}_p = \frac{1}{2}k\omega\left(\frac{\xi_N}{\xi_T} - 1\right)\sum_{n=1} n \tilde{a}_n^2. \tag{S38}$$

Force in the $y$ direction and the corresponding torque

Plugging *Equation S32* into *Equation S31* yields *Equation S40*.

$$f_y = -(\xi_N - \xi_T)\left(\sum_{n=0} a_n(n\omega)\sin(n\omega t - kx)\right)\left(\sum_{n=0} a_n k \sin(n\omega t - kx)\right)^2 \tag{S39}$$

$$f_y = -k^2\omega(\xi_N - \xi_T)\left(\sum_{n=0} na_n \sin(n\omega t - kx)\right)\left(\sum_{i,j} a_i a_j \sin(i\omega t - kx)\sin(j\omega t - kx)\right) \quad \text{(S40)}$$

One may rewrite *Equation S40* in the form of *Equation S41*.

$$f_y = -k^2\omega(\xi_N - \xi_T)\left(\sum_{n,i,j} na_n a_i a_j \sin(n\omega t - kx)\sin(i\omega t - kx)\sin(j\omega t - kx)\right). \quad \text{(S41)}$$

Because we are interested in the time average values of $f_y$, the following terms of *Equation S41* are non-zero:

$$i = 0, j = n$$

$$j = 0, i = n$$

Therefore, *Equation S41* is reduced to *Equations S42 and S43*:

$$f_y = -2a_0 k^2\omega(\xi_N - \xi_T)\sin(-kx)\left(\sum_n na_n^2 \sin^2(n\omega t - kx)\right) \quad \text{(S42)}$$

$$f_y = a_0 k^2\omega(\xi_N - \xi_T)\sin(kx)\left(\sum_n na_n^2(1 - \cos 2(n\omega t - kx))\right). \quad \text{(S43)}$$

Taking the average of *Equation S43*, the average force produced by each segment in the *y* direction is:

$$\tilde{f}_y = \frac{1}{T}\int_0^T f_y dt = (\xi_N - \xi_T)\omega k^2 a_0 \sin(kx)\sum_{n=0} n\tilde{a}_n{}^2. \quad \text{(S44)}$$

The time average of the force produced by each segment of the flagellum in the *y* direction is not constant and is a function of *x*, meaning that the effect of the zeroth harmonic can be seen in the force produced in the *y* direction. Integrating the forces produced by each segment in the *y* direction over the flagellum, the total force produced in the *y* direction becomes 0:

$$\tilde{F}_y = \int_0^L \tilde{f}_y dx = 0. \quad \text{(S45)}$$

However, the magnitude of the force produced in the front half of the sperm is non-zero and equal to the force produced in the rear half of the sperm:

$$\tilde{F}_{yF} = \int_0^{\frac{L}{2}} \tilde{f}_y dx = 2(\xi_N - \xi_T)\omega k a_0 \sum_{n=0} n\tilde{a}_n^2. \quad \text{(S46)}$$

$$\tilde{F}_{yR} = -2(\xi_N - \xi_T)\omega k a_0 \sum_{n=0} n\tilde{a}_n^2. \quad \text{(S47)}$$

We will use two forces for the $\gamma$ ratio, which is required for the Stokeslet description of the micro-swimmer model (Section VIII of Appendix 1). Although the total force produced in the *y* direction is 0, the torque produced by the flagellum is not:

$$\tau_f = \int (x - x_{CM})\tilde{f}_N dx = (\xi_N - \xi_T)\omega k L \tilde{a}_0 \sum_N n\tilde{a}_n^2. \quad \text{(S48)}$$

To find the angular velocity of the sperm ($\tilde{\Omega}$), we need to calculate the torque produced by drag as well:

$$\tau_d = -\int (x - x_{CM})^2 \xi_N \tilde{\Omega} dx = -\xi_N \tilde{\Omega}\frac{L^3}{12}. \quad \text{(S49)}$$

Considering the zero net torque constraint, we calculated the angular velocity of the sperm:

$$\tau_f + \tau_d = 0 \to \tilde{\Omega} = \frac{12(\xi_N - \xi_T)\omega k L \tilde{a}_0}{\xi_N L^3} \sum_N n \tilde{a}_n^2. \tag{S50}$$

Because the curvature of the sperm trajectory is $\tilde{\Omega}\tilde{V}_p^{-1}$ (where $\tilde{\Omega}$ is the angular velocity of the sperm and $\tilde{V}_p$ is the sperm velocity), the curvature of sperm path is:

$$\tilde{\kappa} = \frac{\tilde{\Omega}}{\tilde{V}_p} = \frac{24\tilde{a}_0}{L^2}\left(\frac{\xi_T}{\xi_N}\right) \to L\tilde{\kappa} \propto \frac{\tilde{a}_0}{L}. \tag{S51}$$

We may write the $\gamma$ ratio as follows:

$$\gamma = \frac{\tilde{F}_N}{2\tilde{F}_T} = \frac{2a_0}{L}. \tag{S52}$$

## V Rolling facilitates sperm rheotaxis

Sperm rheotaxis and the subsequent angular velocity imposed by external shear flow, which aligns the sperm in the upstream direction, can be described by an Adler-type equation:

$$\Omega_{Rh} = -A\gamma\sin\theta. \tag{S53}$$

where $\theta$ is the angle between sperm orientation and flow stream $\left(\cos^{-1}\left(\frac{V_P \cdot V_F}{\|V_P\|\|V_F\|}\right)\right)$, $\gamma$ is the shear rate, and $A$ is a constant. Therefore, sperm orientation with respect to the flow evolves with $\Omega_{Rh}$ superimposed with $\Omega_{Asy}$, which is caused by the asymmetry in the beating pattern (*Figure 2—figure supplement 5A*):

$$\dot{\theta} = \Omega_{Rh} + \Omega_{Asy} = -A\gamma\sin\theta + \Omega. \tag{S54}$$

Considering that the angle between sperm orientation and flow while swimming upstream ($\theta_{UP}$) is the angle at which $\dot{\theta} = 0$:

$$\dot{\theta}|_{\theta=\theta_{UP}} = 0 \to \theta_{UP} = \sin^{-1}\left(\frac{\Omega}{A\gamma}\right). \tag{S55}$$

Now, we include the influence of rolling by replacing $\Omega$ with $\Pi(t)\Omega$:

$$\theta_{UP}(t) = \sin^{-1}\left(\frac{\Pi(t)\Omega}{A\gamma}\right) = \Pi(t)\sin^{-1}\left(\frac{\Omega}{A\gamma}\right). \tag{S56}$$

Taking the average value of $\theta_{UP}(t)$ and assuming that $\widetilde{\Pi(t)} \ll 1$:

$$\tilde{\theta}_{UP} = \widetilde{\Pi(t)}\sin^{-1}\left(\frac{\Omega}{A\gamma}\right) \ll \theta_{UP}. \tag{S57}$$

*Equation S57* suggests that rolling results in a significant decrease in $\tilde{\theta}_{UP}$. To see how this decrease in $\tilde{\theta}_{UP}$ facilitates sperm rheotaxis, we calculated the net upstream component of sperm motion with and without the rolling component.

According to the velocity components shown in *Figure 2—figure supplement 5B*, the net upstream velocity is:

$$V_{UP} = V_P\cos(\theta_{UP}) - V_N\sin(\theta_{UP}) - V_F. \tag{S58}$$

where $\theta_{UP}$ includes $\Pi(t)$:

$$V_{UP}(t) = V_P\cos(\Pi(t)\theta_{UP}) - V_N\sin(\Pi(t)\theta_{UP}) - V_F$$

$$V_{UP}(t) = V_P\cos(\theta_{UP}) - \Pi(t)V_N\sin(\theta_{UP}) - V_F \tag{S59}$$

Taking the average value of $V_{UP}(t)$ and assuming that $\widetilde{\Pi(t)} \approx 0$, upstream velocity can be written as follows:

$$\tilde{V}_{UP} = V_P \cos(\tilde{\theta}_{UP}) - \widetilde{\Pi(t)} V_N \sin(\tilde{\theta}_{UP}) - V_F \approx V_P \cos(\tilde{\theta}_{UP}) - V_F. \tag{S60}$$

At high frequencies of rolling, $\tilde{\theta}_{UP}$ approaches 0 (as discussed previously) and therefore,

$$\lim_{\tilde{\theta}_{up} \to 0} \tilde{V}_{UP} = V_P - V_F. \tag{S61}$$

Thus, at the frequent rolling limit, the tangential velocity produced by flagellation is inclined toward upstream of the flow and thus rheotactic behavior is efficient, whereas without the rolling component $V_{\mathrm{UP}}$ is:

$$V_{\mathrm{UP}} = V_P \cos(\theta_{UP}) - V_N \sin(\theta_{UP}) - V_F \approx V_P - V_F - \left( V_N \frac{\Omega}{A\gamma} + \frac{1}{2} V_P \left( \frac{\Omega}{A\gamma} \right)^2 \right), \tag{S62}$$

which is smaller than $V_P - V_F$.

We experimentally measured $\tilde{\theta}_{UP}$ for rolling and non-rolling sperm. For rolling sperm, $\tilde{\theta}_{UP} = 10 \pm 5°$, whereas for non-rolling sperm, $\tilde{\theta}_{UP} = 40 \pm 10°$, which is consistent with our theoretical analysis. By tracking the sperm under flow (*Figure 2—figure supplement 6*), we noticed that the swimming velocity was much higher $(\tilde{V}_{UP} = 60 \pm 10 \mu\mathrm{m/s})$ when the sperm exhibited rolling compared to non-rolling $(\tilde{V}_{UP} = 30 \pm 10 \mu\mathrm{m/s})$.

## VI Noise in the amplitude and phase of the first and higher harmonics

To identify the source of diffusive motion at the center of the circular path, we employed results obtained from the resistive force theory (*Equations S38 and S50*), and included white Gaussian noises in the amplitudes and phases of all harmonics:

$$y(x,t) = \sum_{n=0} \tilde{a}_n (1 + \eta_n(t)) \cos(n\tilde{\omega}t(1 + \xi(t)) - kx) \tag{S63}$$

where $\langle \eta_n(t) \rangle = \langle \xi(t) \rangle = 0$, $\langle \eta_n(t)\eta_{n'}(t') \rangle = D_n \delta_{nn'} \delta(t - t')$, and $\langle \xi(t)\xi(t') \rangle = D_\omega \delta(t - t')$. Inserting the noise terms, we solved the equations of motion for a sperm with arbitrary and constant progressive and angular velocities and noticed that except for the noise in the amplitude of the zeroth harmonics, all other sources of noise did not change the pathway and hence do not result in a diffusive motion of the center (*Figure 3—figure supplement 1*). By contrast, the noise in the amplitude of the zeroth harmonic produces a diffusive component during circular motion. Moreover, the SNR of the zeroth harmonic determines the diffusion coefficient of the center.

## VII Infrequent rolling and intermittent search

Our experimental measurement of the thickness of the layer shown in *Figure 3A*, that is, $\delta$, indicates that it correlates with $\sqrt{t}$, as in a normal diffusion process (*Figure 3—figure supplement 2*). Furthermore, we observed that the sperm with circular diffusive motion at certain ambient viscosities might show infrequent rolling that relocates the center. Measuring the velocity of the center, we demonstrated that relocation happens much faster than diffusion, and subsequently, one can posit the relocations as abrupt interruptions to the diffusion process (*Figure 3—figure supplement 3*). Because sperm circular motion includes a diffusion process with $\delta \propto \sqrt{t}$, and infrequent rolling results in abrupt relocations of the center, we now propose a model to investigate how infrequent rolling increases the area swept by circular motion.

The area swept by the sperm through a sole diffusion process is equivalent to the area of a ring with the thickness $\delta$ and curvature $\kappa$, which is approximately $2\pi\delta\kappa^{-1}$ for $\delta \ll \kappa^{-1}$. Assuming that $\kappa$ is constant in time, the area of the ring increases in time with the square root of time $(\langle A(t) \rangle \propto \sqrt{t})$ (*Figure 3—figure supplement 4*). Now, suppose that the time between $(i-1)$th and $i$th rolling is $T_i$, during which the sperm swims in diffusive circular trajectories. The area that the sperm sweeps after $N$ rolling occurrences is proportional to $\sum_{i=1}^{N} \sqrt{T_i}$, which is larger than the area swept without rolling,

$\sqrt{\sum_{i=1}^{N} T_i}$. This two-phase motion in the sperm is reminiscent of intermittent search algorithms that have been demonstrated to be more efficient than a simple diffusion process.

## VIII The flow field produced by the propulsive and circular components of motility

The flow field produced by the swimmer model in *Figure 4—figure supplement 1* is the superposition of the flow field produced by the propulsive $\left(\vec{f}\right)$ and circular $\left(\vec{f''} \text{ and } -\vec{f''}\right)$ components of motility and their corresponding drags $\left(-\vec{f}, \vec{f'}, \text{ and } -\vec{f'}\right)$. To identify the mechanism of reduction in the far-field drift velocity caused by circular components, we separately simulated the propulsive term with its corresponding drag (*Figure 4—figure supplement 2A*) and the circular term with its corresponding drag (*Figure 4—figure supplement 2B*). The arrows show the normalized vector field, and the magnitude of the flow is represented in color. The flow field produced by the circular terms suppresses the flow field produced by the propulsive term, and thus, reduces the far-field drift velocity.

## IX The analytic solution for the swimmer model

The swimmer model for calculating the far-field drift velocity is shown in *Figure 4—figure supplement 3*. The velocity field imposed by a source point is known as the Stokeslet (*Equation S64*), which is the most fundamental solution for the Stokes equation. Based on the linearity in the Stokes equation, the contribution of the actively swimming sperm on the fluid flow can be described by superimposing the flow fields produced by each point force. For the sake of simplicity, we write the imposed velocity field in three terms, including:

1. The velocity field imposed by $\vec{f}$ and $-\vec{f}$ (*Equation S65*),
2. $\vec{f'}$ and $-\vec{f'}$ (*Equation S66*), and
3. $\vec{f''}$ and $-\vec{f''}$ (*Equation S67*), in which the magnitudes of $\vec{f'}$ and $\vec{f''}$ are equal to $\gamma \vec{f}$.

$$\vec{u}_f = \frac{1}{8\pi\mu}\left\{\frac{\vec{f}}{r} + \frac{\left(\vec{f}\cdot\vec{r}\right)\vec{r}}{r^3}\right\} \tag{S64}$$

$$\vec{u}_{\{f,-f\}} = \vec{u}_f + \vec{u}_{-f} = \frac{p\vec{r}}{8\pi\mu r^3}\left\{-1 + 3\cos^2\varphi\right\} \tag{S65}$$

$$\vec{u}_{\{f',-f'\}} = \frac{1}{8\pi\mu r^3}\left\{\vec{f'}rd\cos\varphi - f'r\sin\varphi\vec{d} + \frac{3rd\cos\varphi}{r^2}f'r\sin\varphi\vec{r}\right\} \tag{S66}$$

$$\vec{u}_{\{f'',-f''\}} = \frac{1}{8\pi\mu r^3}\left\{\vec{f''}rd'\cos(\varphi+\Delta\theta) - f''r\sin(\varphi+\Delta\theta)\vec{d'} - \frac{3}{2}f''d'\sin(2\varphi+2\Delta\theta)\right\}. \tag{S67}$$

Note that $\vec{d}$ is a vector with a magnitude equal to the sperm length in $\hat{d}$ direction and $\vec{d'} = \begin{bmatrix} cos\theta & -sin\theta \\ sin\theta & cos\theta \end{bmatrix}\vec{d}$.

Using *Equations S79, S80, and S81*, the velocity field imposed by the swimmer model is:

$$\vec{u}_T = \frac{p}{8\pi\mu r^2}\left\{A\hat{r} + \left(\bar{\bar{I}}\cos(\varphi) - \bar{\bar{B}}\cos(\varphi+\Delta\theta)\right)\gamma\hat{f} - \left(\bar{\bar{I}}\sin(\varphi) - \bar{\bar{B}}\sin(\varphi+\Delta\theta)\right)\gamma\hat{d}\right\} \tag{S68}$$

with:

$$A = -1 + 3\cos^2\varphi + \frac{3}{2}(\sin(2\varphi) - \sin(2\varphi + 2\Delta\theta))$$

$$\bar{\bar{B}} = \begin{bmatrix} \cos\Delta\theta & -\sin\Delta\theta \\ \sin\Delta\theta & \cos\Delta\theta \end{bmatrix} \text{ and } \bar{\bar{I}} = \begin{bmatrix} 1 & 0 \\ 0 & 1 \end{bmatrix}.$$

To calculate the far-field drift velocity caused by a wall with no-slip boundary condition, we replaced the wall using the mirror image of the swimmer in the boundary. Accordingly, the velocity field imposed on the sperm that causes the far-field drift toward the wall ($U_w$) is $u_T$ with $r = 2h$, where $h$ is the distance between the sperm and the wall. For $\varphi = \pi/2$, $A = -1 + \frac{3}{2}\sin(2\Delta\theta)$ and $\gamma \ll 1$:

$$U_w = \frac{p}{32\pi\mu h^2}\left\{1 - \frac{3}{2}\sin(2\Delta\theta)\right\}. \tag{S69}$$

This equation can also be written as:

$$U_w = U_w^p\left(1 - \frac{3}{2}\sin\left(2\widetilde{\Delta\theta}\right)\right). \tag{S70}$$

where $U_w^p$ is the drift velocity when the circular terms are 0.

Interestingly, at $\widetilde{\Delta\theta}_n = \frac{1}{2}\sin^{-1}\frac{2}{3} \approx 21°$, the swimmer experiences no attraction toward the walls and becomes neutral. Given the relation between $\widetilde{\Delta\theta}$, $\tilde{\kappa}$, and $\tilde{V}_p$, the corresponding neutral curvature is:

$$\tilde{\kappa}_n = \frac{\omega}{4\pi\tilde{V}_p}\sin^{-1}\frac{2}{3}. \tag{S71}$$

## X The effect of the wall on the center of the circular path

Now, using *Equation S70*, we develop a model to calculate the average drift velocity imposed by the wall on the circular path's center in one round of circulation (*Figure 4—figure supplement 4*):

$$\tilde{U}_{wall} = \frac{U_w^p}{N}\sum_{k=0}^{N} z^2\frac{A_k}{h_k^2} \tag{S72}$$

with

$$A_k = -1 + 3\cos^2\varphi_k + \frac{3}{2}(\sin(2\varphi_k) - \sin(2\varphi_k + 2\theta))$$

$$\varphi_k = \frac{\pi}{2} + \frac{2\pi}{N}k; k = 0, 1, \cdots N; \theta = \frac{2\pi}{N}; \frac{R}{z} = \epsilon N$$

where $N$ is the number of beats required to swim a complete circle.

*Equations S73 and S74* are derived by plugging $\varphi_k$ and $\theta$ into *Equation S72* and $h_k$:

$$A_k = -1 + 3\sin^2\left(\frac{2\pi k}{N}\right) + \frac{3}{2}\left(\sin\left(\frac{4\pi(k+1)}{N}\right) - \sin\left(\frac{4\pi k}{N}\right)\right)$$

$$A_k = \frac{1}{2} - \frac{3}{2}\left(\sin\left(\frac{4\pi k}{N}\right) + \cos\left(\frac{4\pi k}{N}\right) - \sin\left(\frac{4\pi(k+)}{N}\right)\right)$$

$$A_k = \frac{1}{2} - \frac{3}{2}\left(\sin\left(\frac{4\pi k}{N}\right)\left[1 - \cos\left(\frac{4\pi}{N}\right)\right] + \cos\left(\frac{4\pi k}{N}\right)\left[1 - \sin\left(\frac{4\pi}{N}\right)\right]\right) \tag{S73}$$

$$h_k = z\left(1 + \epsilon N\left(1 - \cos\left(\frac{2\pi k}{N}\right)\right)\right). \tag{S74}$$

Using *Equation S73* and *Equation S74*, the average far-field drift velocity imposed on the sperm by the wall can be written as follows:

$$\tilde{U}_{wall} = \frac{U_w^p}{N} \sum_{k=0}^{N} \frac{1 - 3\sin\left(\frac{4\pi k}{N}\right)\left[1 - \cos\left(\frac{4\pi}{N}\right)\right] + \cos\left(\frac{4\pi k}{N}\right)\left[1 - \sin\left(\frac{4\pi}{N}\right)\right]}{2\left(1 + \epsilon N\left(1 - \cos\left(\frac{2\pi k}{N}\right)\right)\right)^2}.$$ (S75)

The calculations based on *Equation S75* are shown in *Figure 4—figure supplement 1D*, which indicates that the average drift velocity imposed by the wall on the circular path's center is much lower than the one produced by noise in the amplitude of the zeroth harmonic. Therefore, the circular path is solely evolving with diffusion of the center caused by the noise in the amplitude of the zeroth harmonic, rather than by the far-field influence of the wall.

## XI Near-field interactions and lubrication approximation

To develop a hydrodynamic model and describe sperm near-field interactions with the wall at the lubrication limit, we solved the Stokes equation for a single sperm swimming progressively near a wall with the no-slip boundary condition (*Figure 4—figure supplement 5*). The stress tensor is:

$$\bar{\bar{\sigma}} = -p\bar{\bar{I}} + \mu\left(\nabla u^T + \nabla u\right).$$ (S76)

At distances adequately close to the wall, the contribution of the pressure dominates the viscous term, such that $\sigma \approx -p\bar{\bar{I}}$. Accordingly, the torque exerted on the sperm by the wall can be written as follows:

$$\tau = \int_0^L -p(x - x_{CM})dx.$$ (S77)

Neglecting sperm mass, the net torque applied on the sperm is 0, meaning that the drag torque cancels out the torque exerted by the wall. This constraint gives us the following equation:

$$\dot{\beta} = \frac{d\beta}{dt} = \frac{\alpha}{\xi_N} \frac{\int_0^L p(x - x_{CM})dx}{\int_0^L (x - x_{CM})^2},$$ (S78)

where $\beta$ is the angle of the sperm swimming direction with respect to the wall. To find $\dot{\beta}$, we performed finite element simulations to find the pressure distribution between the sperm and wall, and the results are shown in *Figure 4—figure supplement 5*. Note that the sperm tangential velocity in our simulations was arbitrary and constant, the pressure was found to be linearly correlated to tangential velocity.

## XII The effect of rolling on sperm–wall interaction

In our finite element simulations, we did not include the role of rolling. However, a simple way to include rolling is to include $\Pi(t)$ in the rotation caused by asymmetric flagellar beating. Therefore, we can write $\dot{\beta}$ as follows:

$$\dot{\beta} = g(\beta) + \Pi(t)\Omega.$$ (S79)

To find $\beta_s$ in the presence of rolling, we set $\dot{\beta} = 0$, which yields:

$$\beta_S^*(t) = g^{-1}(-\Pi(t)\Omega)$$

$$\beta_S^*(t) = \Pi(t)g^{-1}(-\Omega) = \Pi(t)\beta_S.$$ (S80)

Taking the average value of $\beta_S(t)$, and assuming that $\widetilde{\Pi(t)} \ll 1$, $\widetilde{\beta_S^*} \ll \beta_S$, suggests that a rolling component in motility significantly decreases the average angle of the sperm with respect to the wall. To find $\beta(t)$ at $\beta \to \widetilde{\beta_S^*}$ limit, we plug $\widetilde{\beta_S}$ into *Equation S79*:

$$\dot{\beta} = g\left(\widetilde{\beta_S^*}\right) + \Pi(t)\Omega \approx \Pi(t)\Omega \tag{S81}$$

where $g\left(\widetilde{\beta_S^*}\right) \approx 0$. Relying upon *Equation S81*, $\beta(t)$ is a triangular function with an offset of $\widetilde{\beta_S^*}$:

$$\beta(t) = \Lambda(t)\Omega + \widetilde{\beta_S^*} \tag{S82}$$

Our experimental measurement of $\beta(t)$ for the rolling sperm swimming along the wall agrees with *Equation S82*. Our results demonstrate that the angle between the rolling sperm and wall is a triangular function with an offset of $\widetilde{\beta_S^*} \sim 10°$ (*Figure 4—figure supplement 6*).

