## [Decision Letter]

**Acceptance summary:**

Mammalian sperm cells achieve locomotion in a liquid environment by beating their flagellum in wave-like pattern. It is well-known that the resulting forward motion is often accompanied by a rolling motion around the cell's longitudinal axis. However, the function of this rolling motion in the context of navigation is not well understood. The authors combine experiments and simulations to show how rolling aids sperm navigation along surfaces and external fluid flows.

**Decision letter after peer review:**

Thank you for submitting your article "Rolling controls sperm navigation in response to the dynamic rheological properties of the environment" for consideration by *eLife*. Your article has been reviewed by 2 peer reviewers, and the evaluation has been overseen by a Reviewing Editor and Anna Akhmanova as the Senior Editor. The reviewers have opted to remain anonymous.

Essential revisions:

1. The conclusion that rolling is "in response to the dynamic rheological properties of the environment", while compelling, could be better supported. To show that sperm swimming *responds* to the rheology of the medium, the authors would first need to track the same sperm in media of different rheologies. Second, it is unclear if the sperm is responding to the rheology, or the swimming merely manifests differently in response to the rheology. Third, the data presented for the rheological characterization (Figure S1) do not support the clean delineation between the PVP media being "viscous" and the PAM media being "viscoelastic": the PVP media are 1-2 orders of magnitude less viscous than the PAM, and in both cases, the viscosity appears to sharply rise at low shear rates (although this is likely noise as it also manifests for the TALP). The PVP media appear to show weak shear thinning, similar to the PAM. Furthermore, only the shear viscosity is presented for both solutions; to adequately characterize viscoelasticity, the authors need to show data for e.g. the first normal stress difference vs shear rate, or the elastic and viscous moduli vs oscillation frequency, for example.

2. The writing in general is dense and highly technical; it would be useful to the reader if the authors could provide broader introduction and Discussion sections that clearly discuss the context of the work, the findings, and the implications of the findings.

3. The authors say in lines 98-99 that an increase in viscoelasticity slightly increased the propulsive velocity. I do not see this in the data.

4. In Figure 1F, what do the tildes indicate? And what are the units of the quantities plotted?

5. Lines 162-164: The authors discuss the diffusion coefficient, but I do not see a quantitative description in the actual data. Can they plot e.g. a MSD vs time?

6. Figure 2A-B: What are the units? What is shown in blue? What is the influence of experimental noise (e.g. in the imaging) on these results?

7. Figure 2G: It is unclear what the points vs the lines are.

8. Figure 2H: The authors claim this is a normal distribution, but it does not appear to be so.

9. Experimental uncertainties are improperly reported throughout. E.g. 38.7 +/- 12.2 should be reported as 40 +/- 10, given the uncertainty in the measurement.

10. What is the influence of thermal diffusion in the results throughout?

11. Line 46: The phrase "To avoid errors arising from studying sperm motility under external fluid flow, …" is unclear – which specific potential "errors" do the authors mean?

12. Line 119: It is not obvious how "Thus, rolling causes progressive motion." follows from the preceding sentences.

13. Line 134: period before equation 1 seems unnecessary?

14. Line 135: How exactly is x defined?

15. Line 142: "(24, 29)(24, 29)(24, 29)(24, 29)"?

16. Line 343 -344: the notation is unclear. Does the tilde indicate a time or ensemble average ? Assuming the former, why is the time average of \Pi still time-dependent ? Shouldn't it read "average of \β_s(t)" in line 344?

17. The authors should carefully check and correct the reference list. The listing of doi 's seems inconsistent and several references are incomplete. For example:

a. Line 524: doi necessary?

b. Line 525: reference incomplete.

c. Line 559: reference incomplete.

d. Line 587: reference incomplete.

e. Line 601: page number seems incorrect.

f. Line 605: page number missing?

g. Line 611: page number seems incorrect.

h. Line 611: page number seems incorrect?

18. Bold font on rhs. of equation S44 should be removed, similar in S51, S52.

19. Please check constant use of multiplicative dots in SI (sometimes lower dots and sometimes centered dots are used).

20. Major point: The definition of eta in equation S65-S68 is either incomplete or incorrect.

21. What exactly do the authors mean by Gaussian noise in S66-67?

22. S67 does not look standard Gaussian white noise, which should be δ correlated. Are the eta(t)'s time correlated or simply random numbers of variance D? In the latter case, the stochastic process eta(t) would be discontinuous?

23. Below equation S77 the authors mention "white Gaussian noise" which seems inconsistent with S67.

24. Also, if eta(t) has a Gaussian distribution, then we do not see how S73 can be satisfied, since the tails of a Gaussian distribution extend to infinity – in particular, when using D=1, as stated below equation S77.

25. Which type of stochastic calculus do the authors use to arrive at S74? It looks like Stratonovich but this is not explicitly stated.

26. Based on the above, we are not convinced that equations S65-S77 are correct.

This part of the SI should be carefully revised and corrected as needed, or omitted.*Reviewer #2 (Recommendations for the authors):*

The authors provide a functional explanation for the observation that mammalian sperm roll around their longitudinal axis: it enables a sperm to continue to move progressively. The authors further sought to provide evidence that rolling responds to the rheological properties of the environment.

The use of imaging to track single sperm and directly observe rolling events is a major strength.

The rheological characterization of the media requires further analysis.

The authors nicely show that rolling enables a sperm to continue to move progressively. However, the conclusion that rolling is "in response to the dynamic rheological properties of the environment", while compelling, is not well supported. The authors tested this hypothesis by studying sperm swimming in a "viscous medium" of TALP + PVP and a "viscoelastic medium" of TALP + PAM, although the rheological characterization of these media does not support this clean delineation.

The observation that sperm rolling has a key functional role broadens understanding of sperm swimming. If the authors could more rigorously support the claim that rolling is modulated in response to the dynamic rheological properties of the environment, then this would broaden understanding of sperm swimming in complex environments during e.g. fertilization.*Reviewer #3 (Recommendations for the authors):*

The wave-like beating patterns of sperm flagella can vary between cells, with some cells exhibiting more symmetric beat patterns than others. To investigate the dependence of rolling motion on beat asymmetry and rheological properties, the author studied bovine sperm motion in a microfluidic device under simple shear flow and within a quiescent reservoir.

The authors' results suggest that rolling becomes suppressed at relatively lower viscosities/viscoelasticity for sperm exhibiting less symmetric beat patterns, with suppression of rolling more sensitive to viscoelasticity than to viscosity. They found that rolling sperm swim on mostly straight trajectories, whereas non-rolling sperm display 2D planar beat forms resulting in circular trajectories.

Building on previous theoretical work, the authors formulate a minimal 2D-projected model that accounts for rolling through a time-dependent prefactor \Pi(t) in equation 4. They find that this model recapitulates their main experimental observations. In particular, their joint theoretical and experimental analysis suggests that, although the rolling component is not required for sperm rheotaxis, it facilitates rheotactic behavior by minimizing the angle between sperm orientation and externally applied fluid flow, thus maximizing the upstream component of sperm motion.

Overall, I find this comprehensive characterization of sperm rolling interesting and it appears to close an existing gap in the literature.

---

## [Author Response]

Essential revisions:1. The conclusion that rolling is "in response to the dynamic rheological properties of the environment", while compelling, could be better supported. To show that sperm swimming responds to the rheology of the medium, the authors would first need to track the same sperm in media of different rheologies.

We thank the reviewers for this helpful comment. To address it, we have conducted new experiments and demonstrated reversibility in the rolling suppression, as we observed reactivation of rolling upon leaving the viscoelastic (1% PAM) or viscous (4% PVP) solutions and entering less viscous and viscoelastic (TALP) solution. This suppression and reactivation of sperm rolling while migrating between TALP and 4% PVP or 1% PAM solutions resulted in transition of sperm swimming behavior between progressive and circular motion. To fully illustrate this point, we have added a new subfigure (Figure 1K) to Figure 1 of this new version of the manuscript, then explained these new findings in the results (page 7, lines 138-149) and further discussed the implications of this observation in the Discussion section (pages 22-23 lines 458-478). Due to the high significance of this reversibility in rolling suppression, we have revised the overall manuscript to reflect this concept throughout.

Second, it is unclear if the sperm is responding to the rheology, or the swimming merely manifests differently in response to the rheology.

Thank you for this helpful comment. We do not have any evidence for the existence of an active mechanism through which sperm “responds” to the rheology of the environment. We replaced “respond” to “in response” throughout the manuscript to clarify. Furthermore, we clearly stated on page 4, lines 72-74 that “sperm swimming behavior manifests differently in response to the rheological properties of the environment”. Unlike small molecules such as caffeine and 4-AP as hyperactivation agonists, there is no evidence of a biochemical response to the PVP or PAM, which are large molecules with molecular weight measured in MDa range, so it is likely that the change in sperm swimming behavior is in response to the rheological properties of the solution.

Third, the data presented for the rheological characterization (Figure S1) do not support the clean delineation between the PVP media being "viscous" and the PAM media being "viscoelastic": the PVP media are 1-2 orders of magnitude less viscous than the PAM, and in both cases, the viscosity appears to sharply rise at low shear rates (although this is likely noise as it also manifests for the TALP). The PVP media appear to show weak shear thinning, similar to the PAM. Furthermore, only the shear viscosity is presented for both solutions; to adequately characterize viscoelasticity, the authors need to show data for e.g. the first normal stress difference vs shear rate, or the elastic and viscous moduli vs oscillation frequency, for example.

Thank you for this very useful comment. We have now measured the viscosity, storage modulus and loss modulus for our polymer-based solutions using a new device which was equipped with large parallel plates (50 mm in diameter). Using such large parallel plates helped us to measure viscosities, as well as G’ and G’’ more precisely. The amplitude sweep was conducted from 0.01% to 100% strain with a 1 Hz angular frequency to identify the linear viscoelastic region. The frequency sweep was performed from 1 to 100 s-1 with a constant 1% strain (within the linear viscoelastic region). The viscosity was measured using the steady shear mode with the shear rate from 0.01 to 100 s-1. Samples were characterized at 37°C. The results of rheological measurements are now available in Supplementary Information, section I (Figure 1—figure supplement 1). These measurements helped us to make more accurate statements in the main text as well.

Based on these new rheological measurements, we noticed that the viscosity of 1% PAM is two orders of magnitude greater than that of 4% PVP. Therefore, both solutions are more viscous than TALP, however, our new rheological measurements also indicate that the storage modulus (G’) of 1% PAM is also two orders of magnitude greater than that of 4% PVP. Therefore, we refer to PVP as the viscous solution and PAM as the viscoelastic solution.

These new measurements helped us to explain the results shown in Figure 1F and G. As demonstrated in Figure 1 G, the propulsive velocity of non-rolling sperm in viscous solution is very low, indicating that increase of viscosity decreases sperm velocity. Further, in PVP, we also observed a similar decay in the frequency of rolling as we increased the concentration (Figure 1F). However, in PAM, we did not observe a decrease in frequency of rolling as we increased the concentration of PAM up to 1%. We also observed two things in 1% PAM: first, that sperm propulsive velocity was measured to be higher than that of the 4% PVP solution, and second, that more than 90% of sperm cells did not show rolling while in 4% PVP only 40% of sperm cells did not roll. Given that the viscosity of 1% PAM is two orders of magnitude higher than that of 4% PVP, these results, based solely on viscosity, seemed contradictory. Therefore, we needed additional parameters beyond viscosity to explain our observations.

To solve this apparent anomaly, we used the additional rheological measurements asked by the reviewers and found that the storage modulus of 1% PAM was two orders of magnitude greater than that of 4% PVP. Therefore, the elasticity of the swimming media plays a major role in the suppression of rolling without loss in the sperm propulsive velocity. This finding was key to our discussion regarding the sensitivity of rolling suppression to viscoelasticity in the previous version, but in this new version this claim is now supported by the rheological measurements.

Including section I of the Supplementary Information, we have also changed Figure 1G according to your comment and our new experiments. We also reflected these findings on pages 5-6 (lines 105-119), pages 23 (lines 469-478), and page 24 (lines 503-509).

2. The writing in general is dense and highly technical; it would be useful to the reader if the authors could provide broader introduction and Discussion sections that clearly discuss the context of the work, the findings, and the implications of the findings.

We thank the reviewers for this comment. According to your comment, we revised the introduction and Discussion sections of the manuscript. Please see pages 2-4 for the revised introduction. Please also see pages 21-23 for the revised discussion.

3. The authors say in lines 98-99 that an increase in viscoelasticity slightly increased the propulsive velocity. I do not see this in the data.

Thank you for calling our attention to this issue. After careful examination of our results, we noticed that our previous claim was not crucial to our discussion, and given the error in our experimental measurements, the slight increase in sperm propulsive velocity was not significant. However, we added Figure 1G to the updated version of the manuscript. This data is important as it demonstrates that suppression of rolling in the viscous solution comes with a significant loss in the propulsive velocity, while suppression of rolling in the viscoelastic solution is not only more efficient but also does not accompany a loss in the sperm propulsive velocity.

4. In Figure 1F, what do the tildes indicate? And what are the units of the quantities plotted?

Thank you for calling our attention to this issue. The tilde signs are now omitted, and the units are added to this figure.

5. Lines 162-164: The authors discuss the diffusion coefficient, but I do not see a quantitative description in the actual data. Can they plot e.g. a MSD vs time?

Thank you for raising this issue. In response, we have plotted the MSD (mean square displacement) of two sperm swimming in diffusive circles and the results are shown in Figure 2E. We noticed that for both sperm, MSD∼t, indicating that the circular motion is indeed diffusive in character.

6. Figure 2A-B: What are the units? What is shown in blue? What is the influence of experimental noise (e.g. in the imaging) on these results?

Thank you for calling our attention to this issue. The units are now added to Figure 2A-B, the blue line in the previous figure was a fitting curve, but it is omitted in this version and the signal itself is shown in blue in Figure 2A.

We discussed the effect of experimental noise in the main text page 8 (lines 161-165). The constant offset in the normalized power spectrum is white noise in our measurement system. Furthermore, the experimental noise coming from our measurement system (image processing) is also included in the peak width around the first harmonic frequency. That is, the peak width centered at ω includes the intrinsic noise originated from flagellar sources, as well as the noise associated with our measurement system.

7. Figure 2G: It is unclear what the points vs the lines are.

Thank you for this comment. To increase clarity, we omitted the points and changed the style of Figure 2G. In HLI graph, the lines are showing the times at which rollings occurred and Π(t) is the defined function.

8. Figure 2H: The authors claim this is a normal distribution, but it does not appear to be so.

Thank you for this comment. We agree with the reviewers, as according to the Kolmogorov–Smirnov test (limiting form, Stephens modification, Marsaglia method, and Lilliefors modification), the distribution shown in Figure 2H is not normal. We omitted this claim in the updated version. Note that simulations performed to obtain sperm trajectories (Figure 2I) was based on the actual distribution of TSR, so we did not change Figure 2I in the updated version.

9. Experimental uncertainties are improperly reported throughout. E.g. 38.7 +/- 12.2 should be reported as 40 +/- 10, given the uncertainty in the measurement.

Thank you for calling our attention to this issue. We carefully checked the experimental errors throughout and modified the reported values.

10. What is the influence of thermal diffusion in the results throughout?

Thank you for this question. In response we have added a new paragraph to the manuscript on page 13 (lines 271-277). Using rough estimations, we argue that thermal fluctuations are negligible in our study. While our estimation is particular to our study, its main idea was borrowed from two previously published studies (ref. 36 and 37).

11. Line 46: The phrase "To avoid errors arising from studying sperm motility under external fluid flow, …" is unclear – which specific potential "errors" do the authors mean?

Thank you. To specify, we revised the sentence according to your comment. The revised sentence on page 3 (lines 50-52) is:

“To avoid errors arising from studying sperm motility under external fluid flow, such as experimental inaccuracies caused by decoupling the effect of flow on sperm motion from active swimming, we employed a multi-step approach.”

12. Line 119: It is not obvious how "Thus, rolling causes progressive motion." follows from the preceding sentences.

Thank you for your comment. We omitted this sentence. Instead, we discussed the reversibility of rolling suppression in the next paragraph (page 7, lines 138-149) and introduced it as another evidence to support our claim that the rolling component is a key contributor to progressive motility.

13. Line 134: period before equation 1 seems unnecessary?

Thank you! We have replaced the period with a colon.

14. Line 135: How exactly is x defined?

Thank you for your comment. We modified the text on page 8 (line 171) to clarify that the x axis was set parallel to the flagellum at its straight-line form.

15. Line 142: "(24, 29)(24, 29)(24, 29)(24, 29)"?

Thank you for calling our attention to this issue. Unfortunately, our reference manager did not work properly. We resolved the issue and deleted this part from the text.

16. Line 343 -344: the notation is unclear. Does the tilde indicate a time or ensemble average ? Assuming the former, why is the time average of \Pi still time-dependent ? Shouldn't it read "average of \β_s(t)" in line 344?

Thank you for this comment. We have modified the text (pages 18-19, lines 377-387) and made the mathematical exposition as clear and easy to follow as possible.

The tilde sign indicates time average, so the tilde sign should be over Π(t), and not just Π.

From the text:

“We then can rewrite β˙ and include Π(t) such that:β˙=g(β)+Π(t)Ω.#(13)

Note that g(β) is the curve in the phase space that corresponds to Ω=0. Insofar as stability occurs at β˙=0,βS*(t)=g−1(−Π(t)Ω)→βS*(t)=Π(t)g−1(−Ω)=Π(t)βS,#(14)in which βS* and βS are the stable points with and without taking rolling into account. Because Π(t)~≪1, the average of βS*(t) is:βS*~=Π(t)~βS≪βS#(15)

Equation 15 suggests that, at the frequent rolling limit where Π(t)~ approaches zero, βS*~ approaches zero as well. Consequently, frequent rollings mitigate the destructive role of Ω in sperm motion along the wall, thereby yielding faster and longer-lasting swimming along the wall (SI, section XII). Because at the frequent rolling limit βS*~ is close to zero, we posit that g(βS*~)≈0, and thus β˙ near the stable point can be written as

β˙=g(βS*~)+Π(t)Ω≈Π(t)Ω.#(16)“

17. The authors should carefully check and correct the reference list. The listing of doi 's seems inconsistent and several references are incomplete. For example:a. Line 524: doi necessary?b. Line 525: reference incomplete.c. Line 559: reference incomplete.d. Line 587: reference incomplete.e. Line 601: page number seems incorrect.f. Line 605: page number missing?g. Line 611: page number seems incorrect.h. Line 611: page number seems incorrect?

Thank you for calling our attention to this issue. Due to the flexibilities offered by *eLife*, we are using the PNAS citation style in this version. We resolved the issue and carefully checked and corrected the reference list. Furthermore, we noticed that the following paper was retracted during the revision process, so we omitted this reference from our reference list.

“Human sperm uses asymmetric and anisotropic flagellar controls to regulate swimming symmetry and cell steering”, (DOI: 10.1126/sciadv.aba5168)

18. Bold font on rhs. of equation S44 should be removed, similar in S51, S52

According to your comment, we removed the bold fonts in equations and mathematical expression throughout the main text and Supplementary Information.

19. Please check constant use of multiplicative dots in SI (sometimes lower dots and sometimes centered dots are used).

According to your comment, we carefully revised the Supplementary Information and removed all multiplicative dots, except for the dot product of two vectors. Accordingly, all dots are simple periods.

20. Major point: The definition of eta in equation S65-S68 is either incomplete or incorrect.21. What exactly do the authors mean by Gaussian noise in S66-67?22. S67 does not look standard Gaussian white noise, which should be δ correlated. Are the eta(t)'s time correlated or simply random numbers of variance D? In the latter case, the stochastic process eta(t) would be discontinuous?23. Below equation S77 the authors mention "white Gaussian noise" which seems inconsistent with S67.24. Also, if eta(t) has a Gaussian distribution, then we do not see how S73 can be satisfied, since the tails of a Gaussian distribution extend to infinity – in particular, when using D=1, as stated below equation S77.25. Which type of stochastic calculus do the authors use to arrive at S74? It looks like Stratonovich but this is not explicitly stated.26. Based on the above, we are not convinced that equations S65-S77 are correct.This part of the SI should be carefully revised and corrected as needed, or omitted.

Thank you for your comments. As suggested, we have chosen to omit this section because it was tangential to our main findings in this work. In the future we plan to revisit this part of this project and to make sure that we can accurately and convincingly address the reviewer’s concerns.